# Inhibitory CCK+ basket synapse defects in mouse models of dystroglycanopathy

Jennifer N Jahncke[1], Daniel S Miller[1], Milana Krush[1], Eric Schnell[2,3], Kevin M Wright[4]*

[1]Neuroscience Graduate Program, Oregon Health & Science University, Portland, United States; [2]Operative Care Division, Portland VA Health Care System, Portland, United States; [3]Anesthesiology and Perioperative Medicine, Oregon Health & Science University, Portland, United States; [4]Vollum Institute, Oregon Health & Science University, Portland, United States

*For correspondence:
wrighke@ohsu.edu

Competing interest: The authors declare that no competing interests exist.

**Abstract** Dystroglycan (Dag1) is a transmembrane glycoprotein that links the extracellular matrix to the actin cytoskeleton. Mutations in *Dag1* or the genes required for its glycosylation result in dystroglycanopathy, a type of congenital muscular dystrophy characterized by a wide range of phenotypes including muscle weakness, brain defects, and cognitive impairment. We investigated interneuron (IN) development, synaptic function, and associated seizure susceptibility in multiple mouse models that reflect the wide phenotypic range of dystroglycanopathy neuropathology. Mice that model severe dystroglycanopathy due to forebrain deletion of *Dag1* or *Pomt2*, which is required for Dystroglycan glycosylation, show significant impairment of $CCK^+/CB_1R^+$ IN development. $CCK^+/CB_1R^+$ IN axons failed to properly target the somatodendritic compartment of pyramidal neurons in the hippocampus, resulting in synaptic defects and increased seizure susceptibility. Mice lacking the intracellular domain of Dystroglycan have milder defects in $CCK^+/CB_1R^+$ IN axon targeting, but exhibit dramatic changes in inhibitory synaptic function, indicating a critical postsynaptic role of this domain. In contrast, $CCK^+/CB_1R^+$ IN synaptic function and seizure susceptibility was normal in mice that model mild dystroglycanopathy due to partially reduced Dystroglycan glycosylation. Collectively, these data show that inhibitory synaptic defects and elevated seizure susceptibility are hallmarks of severe dystroglycanopathy, and show that Dystroglycan plays an important role in organizing functional inhibitory synapse assembly.

## eLife assessment

These **important** findings will be of interest for the study of dystroglycanopathies and in the general area of axon migration and synapse formation. This work provides **convincing** conclusions about how a range of dystroglycan mutations alter CCK interneuron axonal targeting and synaptic connectivity in the forebrain, and seizure susceptibility.

## Introduction

The formation of neural circuits is a multistep process involving proliferation, migration, axon guidance, maturation of neuronal subtypes, and establishment of functional synaptic connections between neurons. The cell adhesion molecule Dystroglycan is widely expressed in muscle and brain. Within the forebrain, Dystroglycan is expressed in neuroepithelial cells, pyramidal neurons, astrocytes, oligodendrocytes, and vascular endothelial cells where it plays important roles in the formation of basement membranes during early brain development (*Colognato et al., 2007*; *Nguyen et al., 2013*; *Nickolls and Bönnemann, 2018*; *Tian et al., 1996*; *Zaccaria et al., 2001*). At later developmental stages,

Dystroglycan is present at multiple synapses, including at photoreceptor ribbon synapses in the retina (*Omori et al., 2012*; *Orlandi et al., 2018*), inhibitory synapses in the cerebellum (*Briatore et al., 2010*; *Briatore et al., 2020*; *Patrizi et al., 2008*), and inhibitory synapses onto pyramidal neurons (*Brünig et al., 2002*; *Lévi et al., 2002*).

Dystroglycan is a central component of the dystrophin-glycoprotein complex (DGC) known primarily for its role in the etiology of neuromuscular diseases including Duchenne muscular dystrophy (DMD), limb-girdle muscular dystrophy (LGMD), and congenital muscular dystrophy (CMD). The gene encoding *Dystroglycan* (*Dag1*) yields two subunits, the extracellular alpha Dystroglycan (α-Dag1) and the transmembrane beta Dystroglycan (β-Dag1). These two subunits are non-covalently bound, allowing Dystroglycan to function as a link between extracellular ligands and cytoskeletal and signaling proteins (*Ervasti and Campbell, 1991*; *Holt et al., 2000*; *Ibraghimov-Beskrovnaya et al., 1992*; *Moore and Winder, 2010*). Extracellular α-Dag1 interacts with multiple proteins in the nervous system through its extensive glycan chains (*Jahncke and Wright, 2023*). Mutations in any of the 19 genes involved in α-Dag1 glycosylation impair Dystroglycan function through reduced ligand binding and leads to a class of congenital muscular dystrophy termed dystroglycanopathy (*Blaeser et al., 2013*). Patients with severe forms of dystroglycanopathy frequently present with structural brain abnormalities and experience seizures and cognitive impairments (*Barresi and Campbell, 2006*; *Muntoni et al., 2011*; *Taniguchi-Ikeda et al., 2016*). Dystroglycanopathy patients with moderate severity can exhibit cognitive impairments even in the absence of identifiable brain malformations, suggesting that Dystroglycan functions at later stages of neural circuit formation such as synapse formation and/or maintenance (*Clement et al., 2008*; *Godfrey et al., 2007*).

The dramatic structural and anatomical phenotypes of global *Dag1* deletion in mice has often precluded analysis of Dystroglycan's synaptic functions (*Myshrall et al., 2012*; *Satz et al., 2008*; *Satz et al., 2010*). Recent studies show that when *Dag1* is selectively deleted from postmitotic pyramidal neurons, neuronal migration and lamination is normal, however CCK$^+$/CB$_1$R$^+$ interneurons (INs) fail to populate the forebrain or form synapses in these mice (*Früh et al., 2016*; *Miller and Wright, 2021*). Furthermore, conditional deletion of *Dag1* from cerebellar Purkinje neurons leads to impaired inhibitory synaptic transmission and a reduction in the number of inhibitory synapses in cerebellar cortex (*Briatore et al., 2020*). These studies establish a role for Dystroglycan function at a subset of inhibitory synapses in the brain, but the critical features of Dystroglycan necessary for these functions, and the relationship between inhibitory synaptogenesis and neurological phenotypes in dystroglycanopathy, remains undefined.

Here, we use multiple mouse models that recapitulate the full range of dystroglycanopathy neuropathology to address several outstanding questions related to the role of Dystroglycan at inhibitory synapses. We find that CCK$^+$/CB$_1$R$^+$ IN axon targeting, synapse formation, and synapse function requires both glycosylation of α-Dag1 and interactions through the intracellular domain of β-Dag1, and that defects in synaptic structure and function is associated with increased seizure susceptibility in mouse models of dystroglycanopathy.

## Results

### Characterizing dystroglycan localization and glycosylation in multiple models of dystroglycanopathy

While conditional deletion of *Dag1* from pyramidal neurons causes a loss of CCK$^+$/CB$_1$R$^+$ IN innervation in the forebrain, this has not been examined in dystroglycanopathy relevant mouse models exhibiting more widespread loss of functional Dystroglycan. We therefore generated five distinct mouse models; three to provide mechanistic insight into Dystroglycan function and two of which model mild dystroglycanopathy (schematized in *Figure 1A*). Since complete loss of *Dag1* results in early embryonic lethality in mice, we generated forebrain-specific conditional knockouts by crossing *Emx1$^{Cre}$* with *Dystroglycan* floxed mice (*Dag1$^{Flox/Flox}$*), to drive recombination in neuroepithelial cells in the dorsal forebrain beginning at embryonic day 10.5 (E10.5) (*Gorski et al., 2002*; *Liang et al., 2012*). We verified the recombination pattern of *Emx1$^{Cre}$* with the mCherry reporter *Rosa26$^{Lox-STOP-Lox-H2B:mCherry}$*. H2B:mCherry signal was present in all excitatory neurons and astrocytes throughout the forebrain (*Figure 1—figure supplement 1A, B and D*) but not microglia or interneurons (*Figure 1—figure supplement 1C and E*).

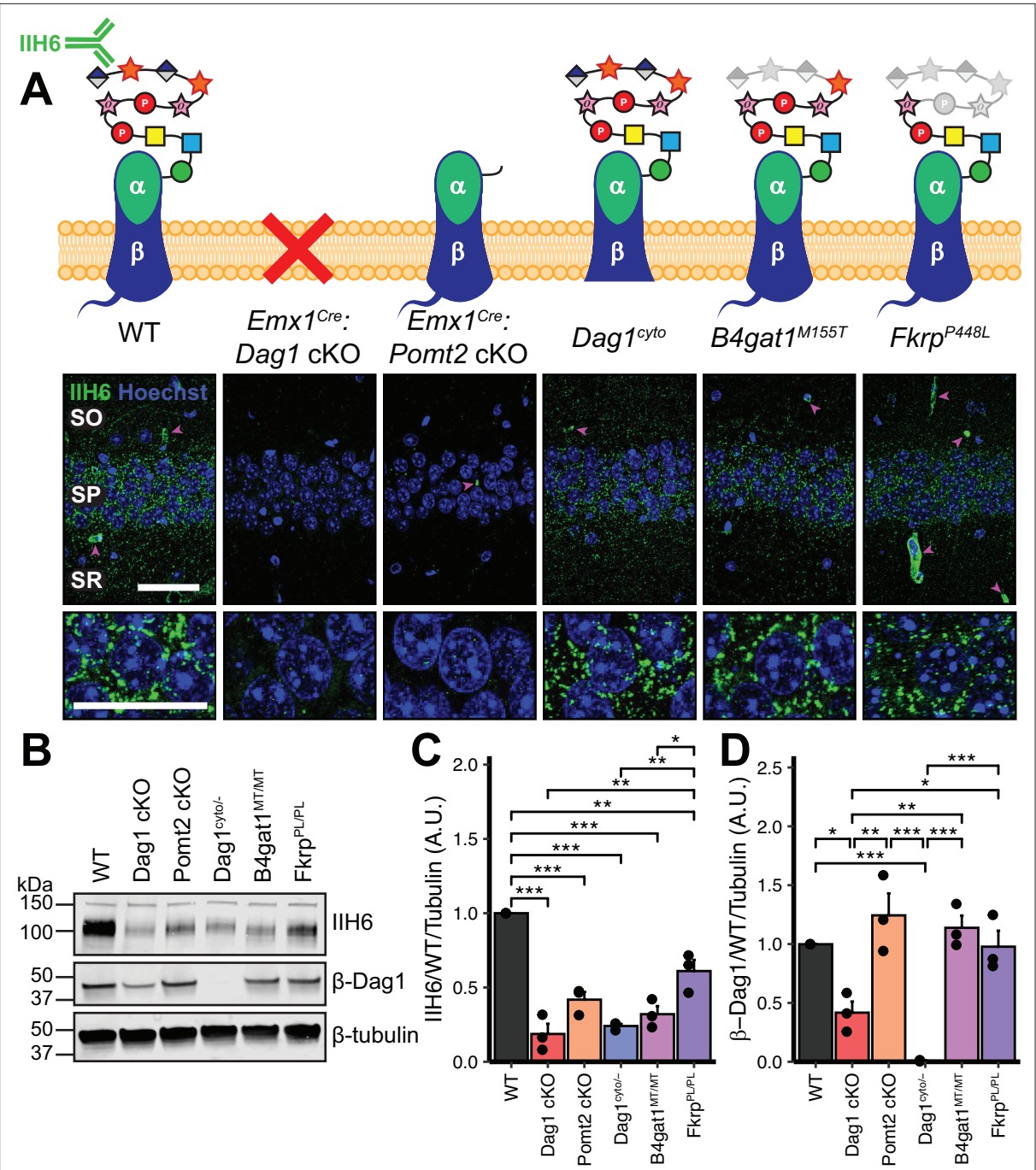

**Figure 1.** Dystroglycan synaptic localization and glycosylation in mouse models of dystroglycanopathy. (**A**) Schematic depiction of Dystroglycan in different mouse models. The IIH6 antibody recognizes the matriglycan repeats on extracellular αDag1. Hippocampal CA1 of P30 mice immunostained for Dystroglycan glycosylation (IIH6, green) and nuclear marker Hoechst (blue) show puncta of glycosylated Dystroglycan localized to the perisomatic region of pyramidal cells and to blood vessels (magenta arrowheads); scale bar = 50 μm. Lower panels show cell bodies in SP; scale bar = 25 μm. CA1 layers: SO, *stratum oriens*; SP, *stratum pyramidale*; SR, *stratum radiatum*. (**B**) WGA-enriched lysates from P0 forebrain were immunoblotted for IIH6, β-Dag1, and β-tubulin. (**C–D**) Quantification of immunoblot in (**B**). Error bars show mean + SEM. See *Supplementary file 1* for Ns.

The online version of this article includes the following source data and figure supplement(s) for figure 1:

**Source data 1.** Raw data for quantification in *Figure 1C, D*.

**Figure supplement 1.** *Emx1Cre* drives recombination in forebrain excitatory neurons and astrocytes, but not interneurons or microglia.

**Figure supplement 2.** Dystroglycan glycosylation is required for synaptic localization.

**Figure supplement 2—source data 1.** Raw data for quantification in *Figure 1—figure supplement 2C, D*.

To model loss of Dystroglycan glycosylation, $Emx1^{Cre}$ was crossed with $Pomt2^{Flox/Flox}$ conditional mice to generate $Emx1^{Cre}$:Pomt2 cKO mice. Pomt2 (protein $O$-mannosyltransferase 2) is a glycosyltransferase that functions in a heterocomplex with Pomt1 to add $O$-mannose at the beginning of the Dystroglycan glycan chain (**Manya et al., 2004**). Without the initial $O$-mannose, no additional sugar moieties can be added to the glycan chain, resulting in near complete loss of Dystroglycan glycosylation. This is lethal embryonically in a global $Pomt2$ knockout, however $Emx1^{Cre}$:Pomt2 cKO mice are viable and survive into adulthood (**Hu et al., 2016**; **van Reeuwijk, 2005**; **Yanagisawa et al., 2007**).

In addition to binding extracellular ligands, Dystroglycan binds cytoskeletal proteins and signals through the intracellular tail of its β-subunit. To determine whether the intracellular domain of Dystroglycan is required for synaptic development and/or function, we examined mice in which one copy of $Dag1$ was deleted, and the other copy lacks the intracellular domain of β-Dag1 ($Dag1^{cyto/-}$). These mice develop muscular dystrophy but show normal neuronal migration and axon guidance in regions throughout the central nervous system where Dystroglycan glycosylation is required (**Lindenmaier et al., 2019**; **Satz et al., 2009**; **Satz et al., 2010**).

To model mild forms of dystroglycanopathy, we examined mice expressing missense mutations in $B4gat1$ (β–1,4-glucuronyltransferase, $B4gat1^{M155T}$) and $Fkrp$ (fukutin related protein, $Fkrp^{P448L}$), two genes required for Dystroglycan glycosylation. $B4gat1^{M155T}$ mice were initially identified in a forward genetic screen and develop mild muscular dystrophy and have diminished ligand binding capacity due to reduced Dystroglycan glycosylation (**Wright et al., 2012**). The $Fkrp^{P448L}$ missense mutation models a mutation found in a patient with dystroglycanopathy (**Brockington et al., 2001**). In mice, the $Fkrp^{P448L}$ mutation leads to reduced glycosylation and mild muscular dystrophy, but no gross brain or eye malformations (**Blaeser et al., 2013**). While it is possible that $Pomt2$, $B4gat1$, and $Fkrp$ could play a role in the glycosylation of proteins other than Dystroglycan, the identity of these proteins has not been described in neurons to date and we did not observe any emergent phenotypes that have not been observed in $Dag1$ mutants (**Gerin et al., 2016**; **Larsen et al., 2017a**; **Larsen et al., 2017b**; **Willer et al., 2014**).

We first examined the pattern of Dystroglycan localization in pyramidal neurons in CA1 of hippocampus in each of the five models by immunostaining adult (P30) mice with the IIH6 antibody, which detects the terminal matriglycan repeats on the glycan chain on α-Dag1 (**Sheikh et al., 2022**; **Yoshida-Moriguchi and Campbell, 2015**). In wild-type (WT) mice, punctate IIH6 immunoreactivity was evident in the somatic and perisomatic compartment of CA1 pyramidal neurons (**Figure 1A**). Immunoreactivity was also present in blood vessels, where $Dag1$ is also expressed (**Durbeej et al., 1998**; **Zaccaria et al., 2001**). Neuronal immunoreactivity was undetectable in $Emx1^{Cre}$:Dag1 cKOs and $Emx1^{Cre}$:Pomt2 cKOs, whereas blood vessel expression was maintained, illustrating the specificity of the conditional deletion (**Figure 1A**). Punctate perisomatic IIH6 immunoreactivity was present in $Dag1^{cyto/-}$, $B4gat1^{M155T/M155T}$, and $Fkrp^{P448L/P448L}$ mice (**Figure 1A**). To assess Dystroglycan localization in $Emx1^{Cre}$:Pomt2 cKOs we used an antibody that recognizes the intracellular C-terminus of β-Dag1. Although immunoreactivity for β-Dag1 was present and elevated above $Emx1^{Cre}$:Dag1 cKO or $Dag1^{cyto/-}$ levels, Dag1 localization did not appear punctate in $Emx1^{Cre}$:Pomt2 cKOs (**Figure 1— figure supplement 2A**). This apparent difference implies that Dag1 glycosylation, and by extension the extracellular interactions that matriglycan mediates, is required for proper Dag1 synaptic localization.

We next prepared WGA-enriched lysate from neonatal (P0) forebrain and immunoblotted for (1) IIH6, to quantify the degree of α-Dag1 glycosylation and (2) β-Dag1, to measure total Dystroglycan protein levels (**Figure 1B**). Dag1 glycosylation was significantly reduced in $Emx1^{Cre}$:Dag1 cKO, $Emx1^{Cre}$:Pomt2 cKO, $Dag1^{cyto/-}$, $B4gat1^{M155T/M155T}$, and $Fkrp^{P448L/P448L}$ mice; however, the reduction in $Fkrp^{P448L/P448L}$ mice was less severe than the other models (**Figure 1C**). The reduction in glycosylation observed in the $Dag1^{cyto/-}$ mice is surprising given that the mutation is restricted to the intracellular domain. $Dystroglycan$ heterozygotes ($Dag1^{+/-}$) show no reduction in IIH6 levels compared to wild-types (data not shown), so the reduction in $Dag1^{cyto/-}$ mice can be presumed to be due to the intracellular deletion. It is possible that the intracellular domain is required for the trafficking of Dystroglycan through the endoplasmic reticulum and/or Golgi apparatus, where Dystroglycan undergoes glycosylation, however additional work is needed to verify this. As expected, β-Dag1 immunoblotting was significantly reduced in $Emx1^{Cre}$:Dag1 cKOs and absent in $Dag1^{cyto/-}$ mice but normal in the glycosylation mutants (**Figure 1D**). The residual β-Dag1 in $Emx1^{Cre}$:Dag1 cKO brain is likely due to

*Dag1* expression in unrecombined cells, such as blood vessels, as well as unrecombined tissue that remained after the forebrain dissection.

We next examined Dag1 protein levels during synaptogenesis using WGA-enriched lysate from P21-P30 hippocampus. As expected, Dag1 glycosylation assessed by IIH6 immunoblotting was severely reduced in *Emx1^Cre^:Dag1* cKOs and *Emx1^Cre^:Pomt2* cKOs but normal in *Dag1^cyto/-^* mice (*Figure 1—figure supplement 2B–C*). Immunoblotting for β-Dag1 showed a significant reduction in *Emx1^Cre^:Dag1* cKOs and *Dag1^cyto/-^* mutants but normal levels in *Emx1^Cre^:Pomt2* cKOs (*Figure 1—figure supplement 2B, D*). Although localization of Dag1 was not punctate in *Emx1^Cre^:Pomt2* cKOs (*Figure 1—figure supplement 2A*), the overall level of β-Dag1 was normal by WGA-enrichment, which enriches for proteins in the plasma membrane (*Figure 1—figure supplement 2B, D*). It therefore remains possible that Dag1 still traffics to the cell surface in *Emx1^Cre^:Pomt2* cKOs but fails to contact presynaptic axons and therefore does not permit synaptogenesis.

## Dystroglycan is required for cortical neuron migration in a glycosylation-dependent manner

In neocortex, *Dag1* expression in radial glia is required for proper migration of neurons, with *Dag1* conditional deletion from neuroepithelial cells or radial glia resulting in Type II lissencephaly (*Moore et al., 2002*; *Pawlisz and Feng, 2011*; *Satz et al., 2008*; *Satz et al., 2010*). This requires proper Dystroglycan glycosylation, but not its expression in neurons (*Chan et al., 2010*; *Holzfeind et al., 2002*; *Hu et al., 2011*; *Wright et al., 2012*). To compare cortical migration across our five models of dystroglycanopathy, we performed immunostaining for the upper layer marker Cux1 (layers II/III-IV) and the deep layer marker Tbr1 (layers III, VI) in P30 somatosensory cortex (*Figure 2A–B*). *Emx1^Cre^:Dag1* cKOs and *Emx1^Cre^:Pomt2* cKOs showed complete cortical dyslamination with 100% penetrance, whereas the cytoplasmic (*Dag1^cyto/-^*) deletion mutants appeared normal (*Figure 2C–D*). *B4gat1^M155T/M155T^* missense mutants showed a migration phenotype only at the cortical midline, while *Fkrp^P448L/P448L^* missense mutants did not show any cortical migration phenotype (*Figure 2C–D*, *Figure 2—figure supplement 1A*). These results indicate that cortical migration depends on Dystroglycan glycosylation but does not require its cytoplasmic domain. Furthermore, taken with the data in *Figure 1B–D*, they illustrate that the severity of the cortical migration phenotype scales with the degree of Dystroglycan hypoglycosylation; *Emx1^Cre^:Dag1* cKOs and *Emx1^Cre^:Pomt2* cKOs model a severe form of dystroglycanopathy (Walker-Warburg Syndrome, Muscle-Eye-Brain disease) and *B4gat1^M155T/M155T^* and *Fkrp^P448L/P448L^* mutants modeling a milder form of the disease.

To further assess the impact of our functional domain mutations, we assessed Laminin localization, the canonical interacting partner of Dystroglycan in the extracellular matrix (ECM), in adult neocortex (*Ibraghimov-Beskrovnaya et al., 1992*). Under WT conditions, Laminin immunoreactivity was evident at the pial surface where Laminin and Dystroglycan interact at the interface between radial glial endfeet and the cortical basement membrane (*Satz et al., 2010*). Laminin was also present in blood vessels, where Dystroglycan-expressing perivascular astrocytes contribute to the maintenance of water homeostasis (*Menezes et al., 2014*). In *Emx1^Cre^:Dag1* cKO cortex, Laminin immunoreactivity at the pial surface was patchy and vascular Laminin showed evidence of the neuronal migration phenotype described in *Figure 2*, *Figure 2—figure supplement 2A*. Laminin immunoreactivity in *Emx1^Cre^:Pomt2* cKO cortex similarly showed a patchy appearance, albeit less severe than *Emx1^Cre^:Dag1* cKOs, along with the evident cortical migration phenotype (*Figure 2—figure supplement 2B*). *Dag1^cyto/-^* mutants, on the other hand, exhibited normal Laminin immunoreactivity both at the pial surface and with regards to vascular organization (*Figure 2—figure supplement 2C*).

## Perisomatic CCK⁺/CB₁R⁺ interneuron targeting requires dystroglycan in a non-cell autonomous manner

CCK⁺/CB₁R⁺ IN innervation is largely absent from the cortex and hippocampus when *Dag1* is deleted selectively from pyramidal neurons using *NeuroD6^Cre^* (*NEX^Cre^*) (*Früh et al., 2016*; *Miller and Wright, 2021*). However, the development and function of CCK⁺/CB₁R⁺ INs has not been examined in mouse models that more broadly lack *Dag1* throughout the CNS and thus more accurately reflect the neuropathology of dystroglycanopathy. We focused our analysis on region CA1 of the hippocampus, as its overall architecture is grossly unaffected in each of our mouse models. Both *Emx1^Cre^:Dag1* cKO

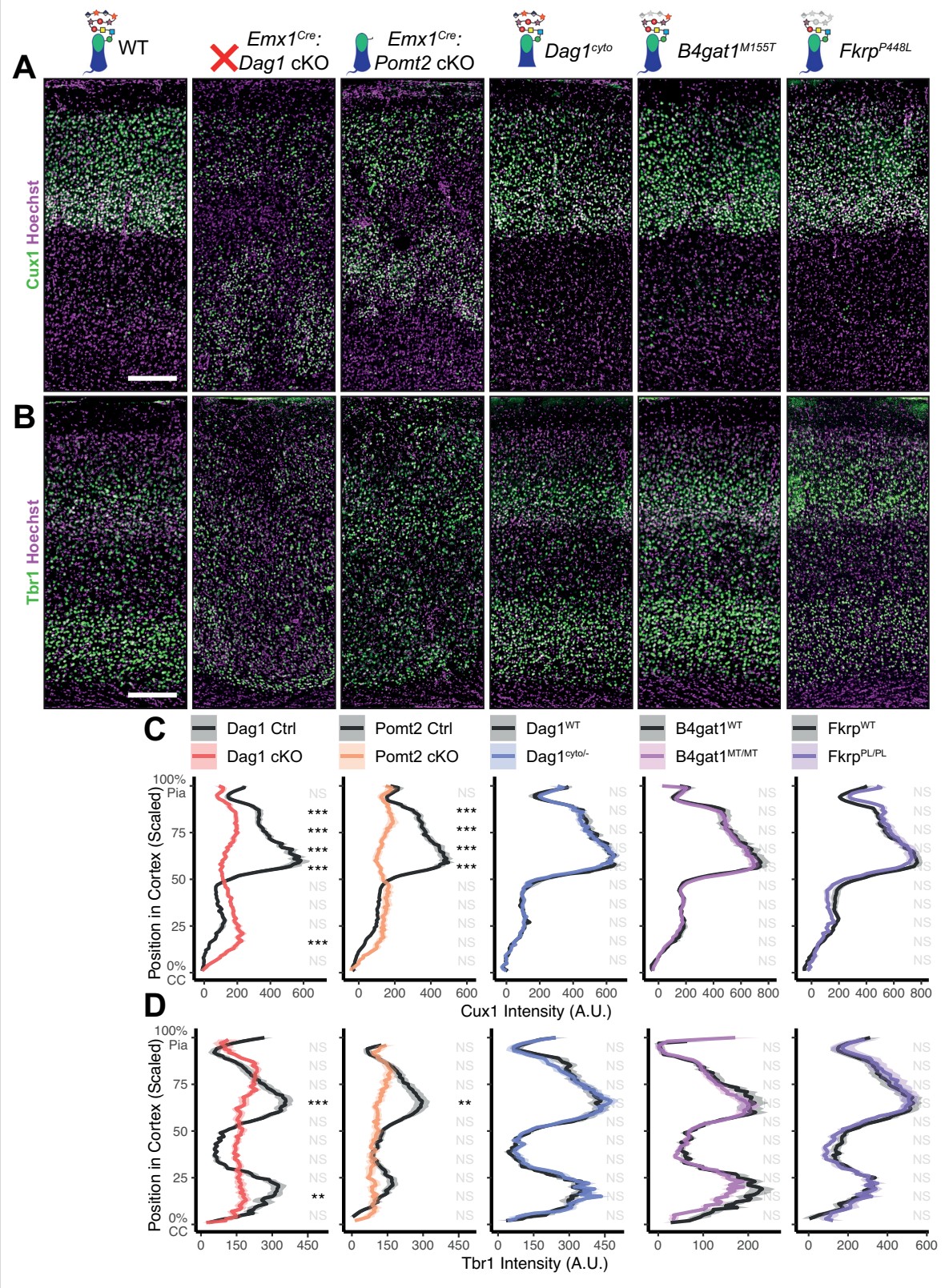

**Figure 2.** Dystroglycan is required for cortical neuron migration in a glycosylation-dependent manner and independent of intracellular interactions. Immunostaining for cortical layer markers (**A**) Cux1 (layers 2–4) and (**B**) Tbr1 (layers 3 and 6) in P30 somatosensory cortex (scale bar = 200 μm). Layer markers are shown in green. Nuclear marker Hoechst is shown in magenta. Quantification of fluorescence intensity of layer markers shown for (**C**) Cux1

*Figure 2 continued on next page*

*Figure 2 continued*

and (**D**) Tbr1. Shaded regions of intensity profile illustrate ± SEM. See **Supplementary file 1** for Ns. *Significance: \*=p < 0.05, \*\*=p < 0.01, \*\*\*=p < 0.001, NS = p ≥ 0.05. CC, corpus callosum; A.U., arbitrary units.*

The online version of this article includes the following source data and figure supplement(s) for figure 2:

**Source data 1.** Raw data for quantification in **Figure 2C**.

**Source data 2.** Raw data for quantification in **Figure 2D**.

**Figure supplement 1.** Cortical migration is disrupted at midline of *B4gat1*$^{M155T/M155T}$ but not *Fkrp*$^{P448L/P448L}$ mutants.

**Figure supplement 2.** Laminin immunoreactivity in adult neocortex appears discontinuous in *Dag1* mutants.

and *Emx1*$^{Cre}$*:Pomt2* cKO mice exhibited a mild granule cell migration phenotype in dentate gyrus (**Figure 3A**, yellow arrows); however, CA1-CA3 showed normal pyramidal neuron organization.

In WT control mice, CCK$^+$/CB$_1$R$^+$ IN axon terminals were abundant throughout the hippocampus, with their highest innervation density in the CA1 pyramidal cell body layer (*stratum pyramidale*, SP) where they form characteristic basket synapses onto pyramidal neurons (**Figure 3A–C**, **Figure 3—figure supplement 1A**). In *Emx1*$^{Cre}$*:Dag1* cKO mice, CCK$^+$/CB$_1$R$^+$ axons were present but failed to target the pyramidal cell layer (**Figure 3A–C**, **Figure 3—figure supplement 1A**), a surprising difference from the phenotype observed in *NeuroD6*$^{Cre}$*:Dag1* cKO mice which lack CCK$^+$/CB1R$^+$ IN innervation entirely (**Früh et al., 2016**; **Miller and Wright, 2021**; **Figure 3—figure supplement 1C**). To confirm the CCK$^+$/CB$_1$R$^+$ IN innervation pattern in the context of widespread *Dag1* deletion, we generated *Nestin*$^{Cre}$*:Dag1* cKO mice. *Nestin*$^{Cre}$, similar to *Emx1*$^{Cre}$, drives *Cre* recombination in forebrain progenitors, however *Emx1*$^{Cre}$ recombination begins around E10.5 and *Nestin*$^{Cre}$ recombination begins around E11.5 (**Liang et al., 2012**; **Tronche et al., 1999**). *Nestin*$^{Cre}$*:Dag1* cKOs showed the same CCK$^+$/CB$_1$R$^+$ axon targeting phenotype as *Emx1*$^{Cre}$*:Dag1* cKOs (**Figure 3—figure supplement 1B**), further suggesting that the observed *Emx1*$^{Cre}$*:Dag1* cKO phenotype faithfully models dystroglycanopathy neuropathology.

It was previously reported that the lack of CCK$^+$/CB$_1$R$^+$ IN innervation of CA1 pyramidal neurons observed in *NeuroD6*$^{Cre}$*:Dag1* cKOs was accompanied by reduced numbers of CCK$^+$/CB$_1$R$^+$ INs (**Miller and Wright, 2021**). We therefore sought to quantify CCK$^+$/CB$_1$R$^+$ IN cell density in *Emx1*$^{Cre}$*:Dag1* cKOs using both NECAB1 and NECAB2 antibodies (**Miczán et al., 2021**). As NECAB1 is expressed by both CCK$^+$/CB$_1$R$^+$ INs and PV$^+$ INs, we performed immunolabeling for both NECAB1 and PV and quantified the density of NECAB1$^+$;PV$^-$ cell bodies in CA1, finding no difference between *Emx1*$^{Cre}$*:Dag1* controls and *Emx1*$^{Cre}$*:Dag1* cKOs (**Figure 3—figure supplement 2A–B**). To confirm this, we also quantified the density of NECAB2$^+$ cell bodies in CA1, again finding no difference between genotypes (**Figure 3—figure supplement 2C, E**). Thus, the observed change in CCK$^+$/CB$_1$R$^+$ IN axon targeting of CA1 pyramidal cells is not due to a reduction in cell numbers, but rather a failure to innervate the appropriate compartment.

*Emx1*$^{Cre}$*:Pomt2* cKO mice fully phenocopied the aberrant *Emx1*$^{Cre}$*:Dag1* cKO CB$_1$R$^+$ immunoreactivity pattern (**Figure 3A–C**), demonstrating that proper CCK$^+$/CB$_1$R$^+$ IN basket axon targeting requires Dystroglycan glycosylation. Both the *B4gat1* and *Fkrp* mutants showed a normal distribution of CB$_1$R$^+$ axon targeting to the somatodendritic compartment of CA1 neurons, but with reduced CB$_1$R intensity in SP (**Figure 3A–C**). The cytoplasmic domain of Dystroglycan also plays a role in the appropriate targeting of CB$_1$R immunoreactive axons, as the distribution of axons was perturbed in the *Dag1*$^{cyto/-}$ mutants, although with an intermediate phenotype in which the upper portion of SP appeared normal while the lower portion of SP showed loss of selective CB$_1$R$^+$ axon targeting (**Figure 3A–C**).

Due to the axonal targeting defect in the CCK$^+$/CB$_1$R$^+$ IN population, we next examined the parvalbumin (PV) population of interneurons in the hippocampus, as these cells also form perisomatic basket cell synapses on to CA1 pyramidal cells. There was no significant difference in the number of PV$^+$ INs in CA1 of *Emx1*$^{Cre}$*:Dag1* cKO mice and littermate controls (**Figure 3—figure supplement 2D and F**). Furthermore, the distribution of PV$^+$ IN axons showed normal targeting to SP in all mouse models (**Figure 3—figure supplement 3A–B**), indicating that the axon targeting phenotype is specific to the CCK$^+$/CB$_1$R$^+$ IN population. Interestingly, *Emx1*$^{Cre}$*:Dag1* cKO mice exhibited a slight increase in PV intensity in SP, perhaps indicating that there is a degree of compensation (**Figure 3—figure supplement 3A–B**).

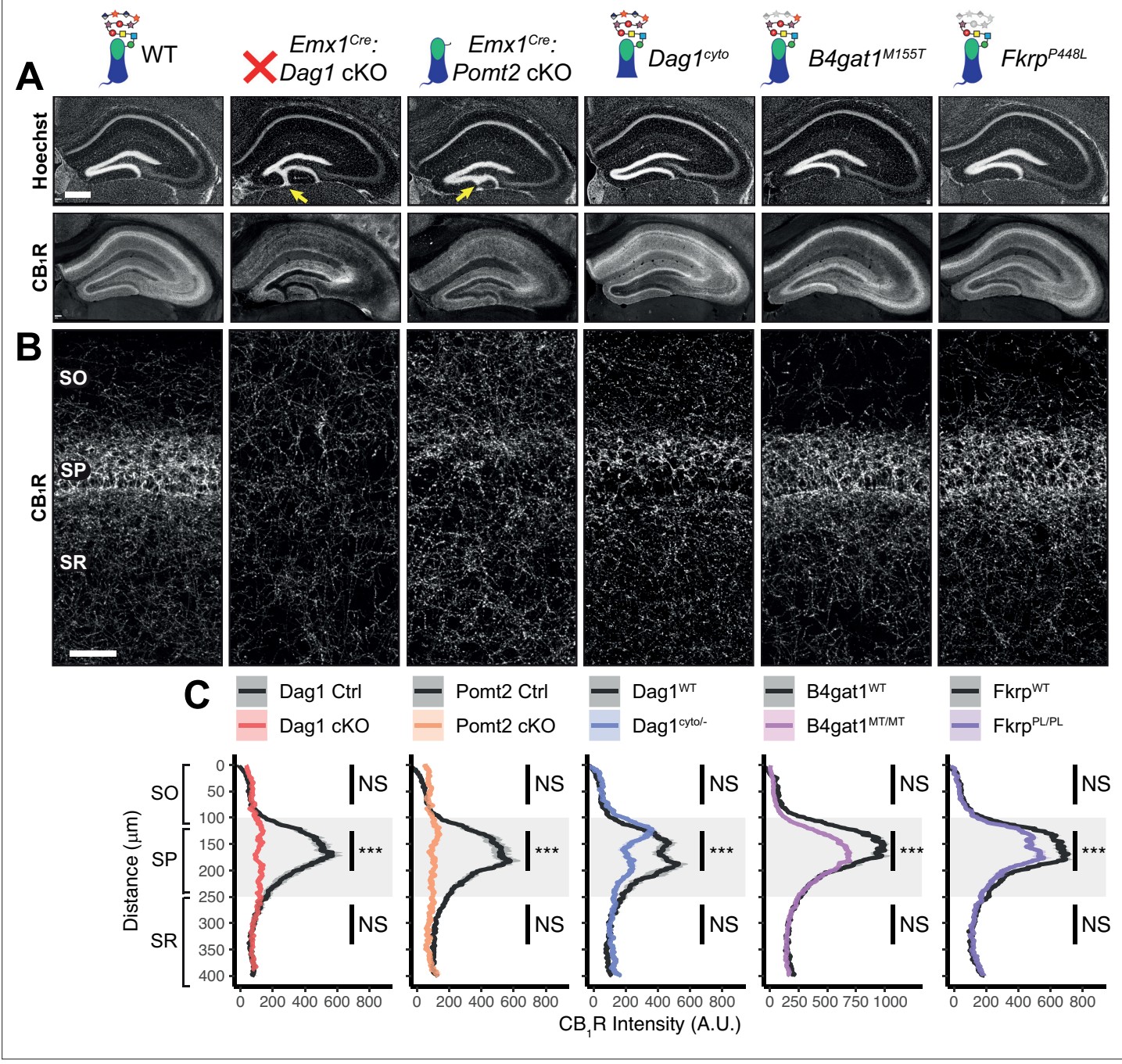

**Figure 3.** *Dag1* is required for CCK+/CB$_1$R+ basket IN perisomatic axon targeting in *stratum pyramidale* of hippocampal CA1-3. (**A**) Nuclear marker Hoechst (upper panels) shows hippocampal morphology. Granule cell migration is disrupted in dentate gyrus of *Emx1^Cre^:Dag1* cKOs and *Emx1^Cre^:Pomt2* cKOs (yellow arrows). CA1-3 gross morphology is normal in all models. CB$_1$R immunostaining (lower panels) shows abnormal CCK+/CB$_1$R+ basket interneuron targeting in CA1-3 to varying degrees across models (scale bar = 400 µm). (**B**) Higher magnification view of CB$_1$R immunostaining in CA1 (scale bar = 50 µm). (**C**) Quantification of CA1 CB$_1$R fluorescence intensity profile. Shaded regions of intensity profile illustrate ± SEM. Gray region highlights SP. See ***Supplementary file 1*** for Ns. *Significance: *=p < 0.05, **=p < 0.01, ***=p < 0.001, NS = p ≥ 0.05. A.U., arbitrary units; SO, stratum oriens; SP, stratum pyramidale; SR, stratum radiatum.*

The online version of this article includes the following source data and figure supplement(s) for figure 3:

**Source data 1.** Raw data for quantification in ***Figure 3C***.

**Figure supplement 1.** CCK+/CB$_1$R+ IN axon targeting phenotypes in hippocampal CA1 of various *Dag1* cKOs.

**Figure supplement 2.** CCK+/CB$_1$R+ IN cell numbers are unchanged in *Emx1^Cre^:Dag1* cKOs.

*Figure 3 continued on next page*

*Figure 3 continued*

**Figure supplement 2—source data 1.** Raw data for quantification in *Figure 3—figure supplement 2B, E, and F*.

**Figure supplement 3.** Parvalbumin$^+$ basket INs do not require *Dag1* for proper axon targeting in hippocampal CA1.

**Figure supplement 3—source data 1.** Raw data for quantification in *Figure 3—figure supplement 3B*.

**Figure supplement 4.** Altered CB$_1$R expression in cortex and basolateral amygdala of *Dag1* and *POMT2* mutants.

Notably, CB$_1$R expression was also abnormal in brain regions outside of the hippocampus. In somatosensory cortex, CB$_1$R immunostaining reflected the dyslamination phenotype in both the *Emx1$^{Cre}$:Dag1* cKO and *Emx1$^{Cre}$:Pomt2* cKO mice throughout neocortex and the *B4gat1$^{M155T/M155T}$* mice at midline, while it appeared normal in cortex of *Dag1$^{cyto/-}$* and *Fkrp$^{P448L/P448L}$* mice (*Figure 3—figure supplement 4A*). CB$_1$R staining was also reduced and disorganized in the basolateral amygdala (BLA) in *Emx1$^{Cre}$:Dag1* cKO and *Emx1$^{Cre}$:Pomt2* cKO mice (*Figure 3—figure supplement 4B*). Interestingly, CB$_1$R immunostaining in the inner molecular layer (IML) of dentate gyrus appears normal in all mutants (*Figure 3A*). In the IML, CB$_1$R is present in excitatory mossy cell axons targeting dentate granule cells whereas in both cortex and amygdala CB$_1$R expression is restricted to GABAergic interneurons (*Földy et al., 2006*; *Katona et al., 2001*; *Monory et al., 2015*). Therefore, glycosylated Dystroglycan instructs the development of inhibitory CB$_1$R$^+$ interneuron populations in multiple brain regions.

## Dystroglycan is required for CCK$^+$/CB$_1$R$^+$ interneuron axon targeting during early postnatal development

During early postnatal development, CCK$^+$/CB$_1$R$^+$ IN axons undergo a dramatic laminar rearrangement, progressing from more distal localization amongst pyramidal cell dendrites, to eventually target pyramidal neuron cell bodies in the hippocampus (*Miller and Wright, 2021*; *Morozov et al., 2009*; *Morozov and Freund, 2003a*; *Morozov and Freund, 2003b*). We examined the developmental time course of CCK$^+$/CB$_1$R$^+$ IN axon targeting in our *Emx1$^{Cre}$:Dag1* cKO mice beginning at P5, when the axons are first readily identifiable (*Berghuis et al., 2007*; *Eggan et al., 2010*; *Mulder et al., 2008*; *Vitalis et al., 2008*). At P5 in *Emx1$^{Cre}$:Dag1* control mice, CCK$^+$/CB$_1$R$^+$ axons were initially concentrated in the *stratum radiatum* (SR) of the hippocampus (*Figure 4A–C*). Between P10 and P30, the CCK$^+$/CB$_1$R$^+$ axons underwent developmental reorganization, with reduced innervation of SR coinciding with a progressive increase in innervation of SP. In contrast, overall CCK$^+$/CB$_1$R$^+$ innervation was initially reduced in the hippocampus of mutant *Emx1$^{Cre}$:Dag1* cKO mice at P5, and these axons failed to undergo laminar reorganization as they developed (*Figure 4A–C*). By P30, after IN synapse formation and targeting are largely complete in control mice, the density of CCK$^+$/CB$_1$R$^+$ axons in *Emx1$^{Cre}$:Dag1* cKO mice was uniform across all hippocampal lamina (*Figure 4A–C*). Therefore, Dystroglycan plays a critical developmental role during the first two postnatal weeks, for the proper laminar distribution and perisomatic targeting of CCK$^+$/CB$_1$R$^+$ IN axons in the hippocampus.

## CCK$^+$/CB$_1$R$^+$ IN synapse formation requires postsynaptic glycosylated dystroglycan

Given the perturbed distribution of CCK$^+$/CB$_1$R$^+$ IN axons in the hippocampus, we next wanted to determine whether the remaining CCK$^+$/CB$_1$R$^+$ IN axons were capable of forming synapses in dystroglycanopathy models. Using VGAT as a marker of inhibitory presynaptic terminals, we saw no difference in total VGAT puncta density in SP in any of the mouse models, indicating that the total number of inhibitory synapses is normal (*Figure 5B–D*). Immunostaining for CB$_1$R showed a significant decrease in CB$_1$R in SP of *Emx1$^{Cre}$:Dag1* cKO, *Emx1$^{Cre}$:Pomt2* cKO, *B4gat1$^{M155T/M155T}$*, and *Fkrp$^{P448L/P448L}$* mutants, but not *Dag1$^{cyto/-}$* mutants (*Figure 5E*). This suggests that the difference in CB$_1$R$^+$ axon distribution described in SP of *Dag1$^{cyto/-}$* mutants in *Figure 3A–C* likely reflects a change in CCK$^+$/CB$_1$R$^+$ IN axon targeting but not synapse formation, whereas *Emx1$^{Cre}$:Dag1* cKO and all three glycosylation mutants exhibit a reduction in CCK$^+$/CB$_1$R$^+$ IN axon targeting and synapse number in SP. It should be noted that the data in *Figure 3A–C* reflects axonal CB$_1$R intensity across all hippocampal layers, whereas the quantification in *Figure 5E* reflects the density of axonal swellings within SP. These data therefore suggest that there is an overall reduction in CB$_1$R intensity in SP of *Dag1$^{cyto/-}$* mutants that does not influence the number of CB$_1$R$^+$ axonal swellings. In contrast to CCK$^+$/CB$_1$R$^+$ INs, the PV$^+$ population of basket interneurons showed no change in puncta density in SP in any of the models (*Figure 5—figure*

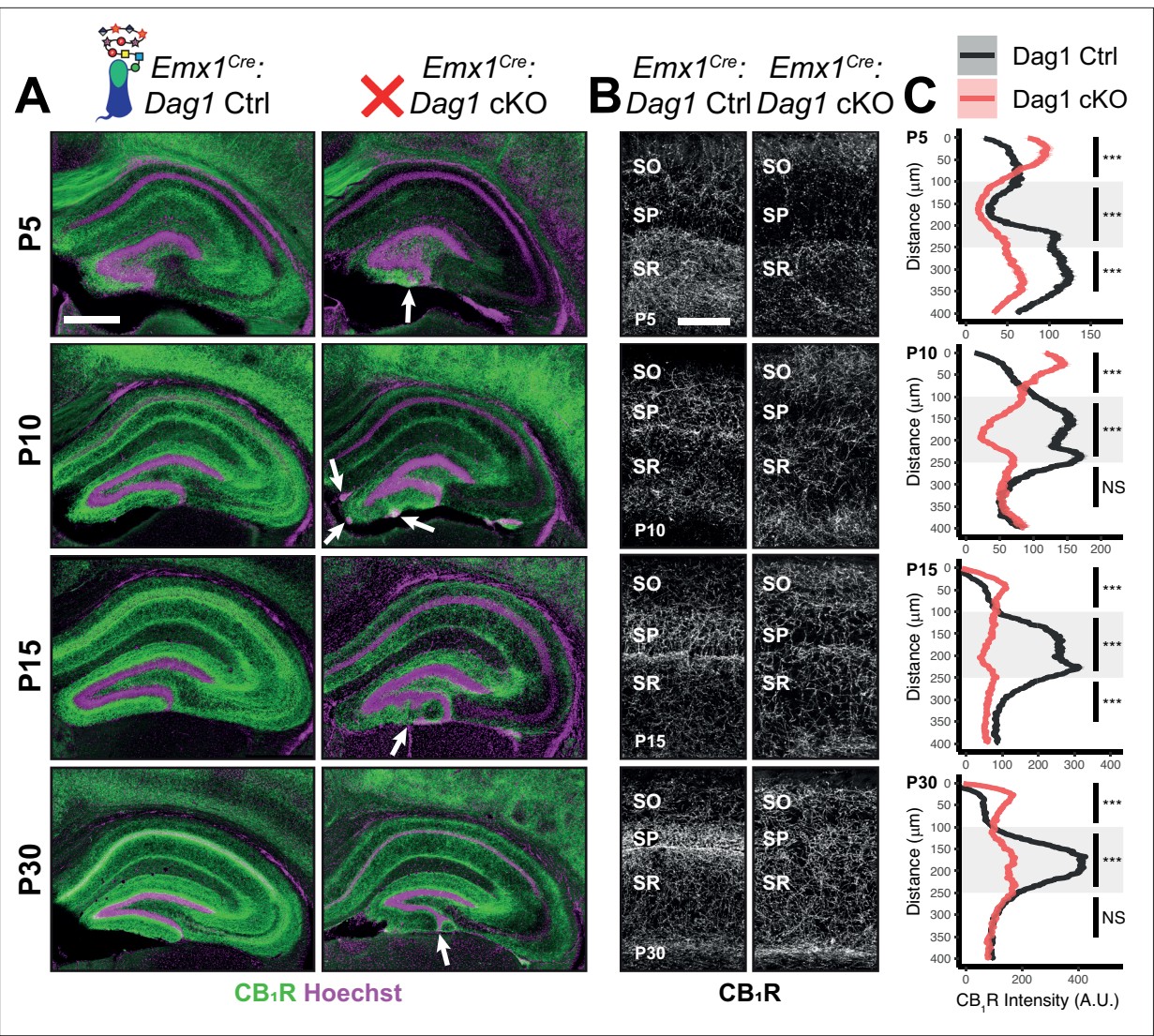

**Figure 4.** *Dag1* is required for CCK$^+$/CB$_1$R$^+$ IN axon targeting during early postnatal development. (**A**) Immunostaining for CB$_1$R$^+$ axon terminals (green) in the hippocampus of *Emx1$^{Cre}$:Dag1* controls (left) and cKOs (right) at ages P5-P30. Nuclear marker Hoechst is shown in magenta. White arrowheads indicate migration errors in dentate granule cells *Emx1$^{Cre}$:Dag1* cKO mice. Scale bar = 500 μm. (**B**) Higher magnification images of CB$_1$R$^+$ axon terminals in CA1 of *Emx1$^{Cre}$:Dag1* controls (left) and cKOs (right) at ages P5-P30. Scale bar = 100 μm. (**C**) Quantification of CB$_1$R fluorescence intensity profile in CA1. Shaded regions of intensity profile illustrate ± SEM. Gray region highlights SP. See ***Supplementary file 1*** for Ns. *Significance: \*=p < 0.05, \*\*=p < 0.01, \*\*\*=p < 0.001, NS = p ≥ 0.05. A.U., arbitrary units; SO, stratum oriens; SP, stratum pyramidale; SR, stratum radiatum.*

The online version of this article includes the following source data for figure 4:

**Source data 1.** Raw data for quantification in ***Figure 4C***.

supplement 2A–E; analysis of VGAT, CB$_1$R, and PV densities in SO and SR included in ***Figure 5—figure supplement 1A–B*** and ***Figure 5—figure supplement 3A***.)

To better approximate the extent of basket synapse formation, we quantified the co-localization between VGAT and CB$_1$R or PV. In SP, the percent of CB$_1$R puncta co-localized with VGAT was reduced in the same models that showed a reduction in CB$_1$R density (*Emx1$^{Cre}$:Dag1* cKO, *Emx1$^{Cre}$:Pomt2* cKO, *B4gat1$^{M155T/M155T}$*, and *Fkrp$^{P448L/P448L}$* mutants) but not *Dag1$^{cyto/-}$* mutants (***Figure 5C and F***), suggesting that CCK$^+$/CB$_1$R$^+$ INs require postsynaptic glycosylated Dystroglycan in order to form synapses whereas the cytoplasmic domain is required for axon targeting but not synapse formation.

Interestingly, the percent of PV co-localized with VGAT increased in the SP of *Emx1$^{Cre}$:Dag1* cKO and *Emx1$^{Cre}$:Pomt2* cKO mice, with no change in any of the other models (***Figure 5—figure supplement 2C, E***; analysis of co-localization in SO and SR included in ***Figure 5—figure supplement 1C***,

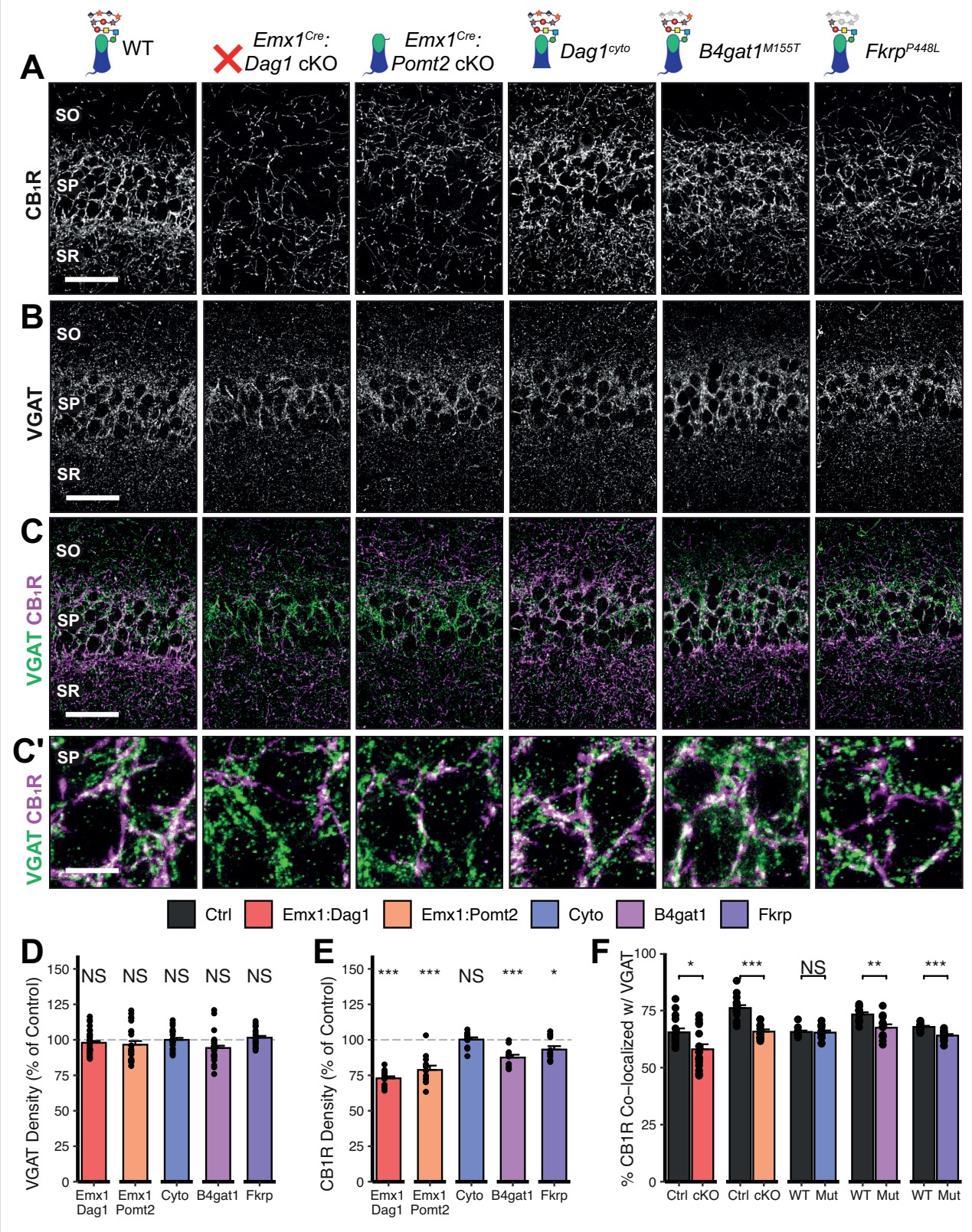

**Figure 5.** *Dag1* and *Pomt2* cKOs exhibit impaired CB1R+ basket synapse formation in *stratum pyramidale* of hippocampal CA1. P30 coronal sections immunostained for (**A**) CB₁R and (**B**) VGAT in hippocampal CA1; merged image in (**C**) shows CB₁R in magenta and VGAT in green. (**A–C**) Scale bar = 50 μm. (Higher magnification view of SP in (**C'**); scale bar = 10 μm.) (**D**) Quantification of VGAT puncta density in SP expressed as a percent of control. (**E**) Quantification of CB₁R puncta density in SP expressed as a percent of control. (**F**) Quantification of co-localization between VGAT and CB₁R in SP to

*Figure 5 continued on next page*

*Figure 5 continued*

estimate putative $CB_1R^+$ basket cell synapse formation. Error bars show mean + SEM. (For quantification of puncta densities and co-localization in SO and SR see *Figure 5—figure supplement 1*.) See *Supplementary file 1* for Ns. *Significance:* *=p < 0.05, **=p < 0.01, ***=p < 0.001, NS = p ≥ 0.05. SO, stratum oriens; SP, stratum pyramidale; SR, stratum radiatum.

The online version of this article includes the following source data and figure supplement(s) for figure 5:

**Source data 1.** Raw data for quantification in *Figure 5D, E*.

**Source data 2.** Raw data for quantification in *Figure 5F*.

**Figure supplement 1.** Extended quantification of images in *Figure 5A–C*.

**Figure supplement 2.** *Dag1* and *Pomt2* cKOs exhibit increased PV$^+$ basket synapse formation in *stratum pyramidale* of hippocampal CA1.

**Figure supplement 2—source data 1.** Raw data for quantification in *Figure 5—figure supplement 2D*.

**Figure supplement 2—source data 2.** Raw data for quantification in *Figure 5—figure supplement 2E*.

**Figure supplement 3.** Extended quantification of images in *Figure 5—figure supplement 2A–C*.

*Figure 5—figure supplement 3B*). It is possible that the reduction in inhibitory $CCK^+/CB1R^+$ synapses prompts homeostatic compensation through an increase in $PV^+$ synapses. Alternatively, this may reflect competition between $CCK^+/CB_1R^+$ and $PV^+$ INs for physical space on the perisomatic region of pyramidal cells, with the decrease in $CCK^+/CB_1R^+$ synapses in *Emx1$^{Cre}$:Dag1* cKOs and *Emx1$^{Cre}$:Pomt2* cKOs allowing additional $PV^+$ IN synapses to form.

## $CCK^+/CB_1R^+$ interneuron basket synapse function is dependent on dystroglycan function

Perisomatic inhibitory basket cell synapses powerfully control activity in the hippocampal circuit (*Freund and Katona, 2007*). Previous studies in *NeuroD6$^{Cre}$:Dag1* cKO mice, in which $CCK^+/CB_1R^+$ INs are absent, demonstrated reduced inhibitory synaptic function (*Früh et al., 2016*). In the current study, however, $CCK^+/CB_1R^+$ INs are present but mistargeted. Thus, we wanted to determine whether with the changes in $CCK^+/CB_1R^+$ basket synapse localization in our mouse models were associated with altered inhibitory synaptic function. $CCK^+/CB_1R^+$ IN basket cells can be selectively activated by muscarinic receptor activation, which increases the rate of spontaneous inhibitory post-synaptic currents (sIPSCs) in nearby pyramidal cells (*Früh et al., 2016*; *Nagode et al., 2014*). Thus, to assay function at $CCK^+/CB_1R^+$ IN synapses, we performed whole cell patch clamp electrophysiology from CA1 pyramidal neurons in slices from control and mutant mice. After recording 5 minutes of baseline sIPSCs, the cholinergic receptor agonist Carbachol (CCh) was added to the bath and an additional 5 minutes of sIPSCs were recorded. While both *Emx1$^{Cre}$:Dag1* control and *Emx1$^{Cre}$:Dag1* cKO cells displayed a CCh-mediated change in sIPSC frequency, this response was dramatically attenuated in *Emx1$^{Cre}$:Dag1* cKOs mice compared to *Emx1$^{Cre}$:Dag1* control mice (*Figure 6A–C*). Furthermore, in *Emx1$^{Cre}$:Dag1* controls, 19/21 cells (90.5%) responded to CCh application (defined as a>20% increase in sIPSC frequency), whereas only 13/22 cells (59.1%) responded in *Emx1$^{Cre}$:Dag1* cKOs (*Figure 6—figure supplement 1B*). Proper Dystroglycan glycosylation was also required for $CCK^+/CB_1R^+$ IN synapse function, as *Emx1$^{Cre}$:Pomt2* cKO mice exhibited the same phenotype as *Emx1$^{Cre}$:Dag1* cKOs: a reduced response to CCh overall, and a reduced proportion of responsive cells (*Figure 6A–C*, *Figure 6—figure supplement 1B*). CCh also increased the mean sIPSC amplitude in each of the controls (*Figure 6—figure supplement 1A*), which may reflect an increased contribution of larger-amplitude action potential-mediated perisomatic events elicited by CCh (*Früh et al., 2016*; *Nagode et al., 2014*). Consistent with the decreased function of $CCK^+/CB_1R^+$ IN synapses, a CCh-mediated change in sIPSC amplitude was also absent in each of these models (*Figure 6—figure supplement 1A*). Together, these data indicate that the altered perisomatic $CCK^+/CB_1R^+$ IN synaptic localization in CA1 is associated with a functional deficit in synaptic signaling.

*Dag1$^{cyto/-}$* mutants also had a dramatically attenuated sIPSC response to CCh compared to WT controls (*Figure 6C*). Notably, even baseline sIPSC frequency was reduced in *Dag1$^{cyto/-}$* mutants (2.27±1.70 Hz) compared to WT controls (4.46±2.04 Hz, *p*=0.002), whereas baseline sIPSC frequencies appeared normal in all other mutants when compared to their respective controls. Together with the finding that these mutants contain a normal number of $CCK^+/CB_1R^+$ basket synapses (as measured using immunohistochemistry; *Figure 5A–D*), these results indicate that the cytoplasmic domain of

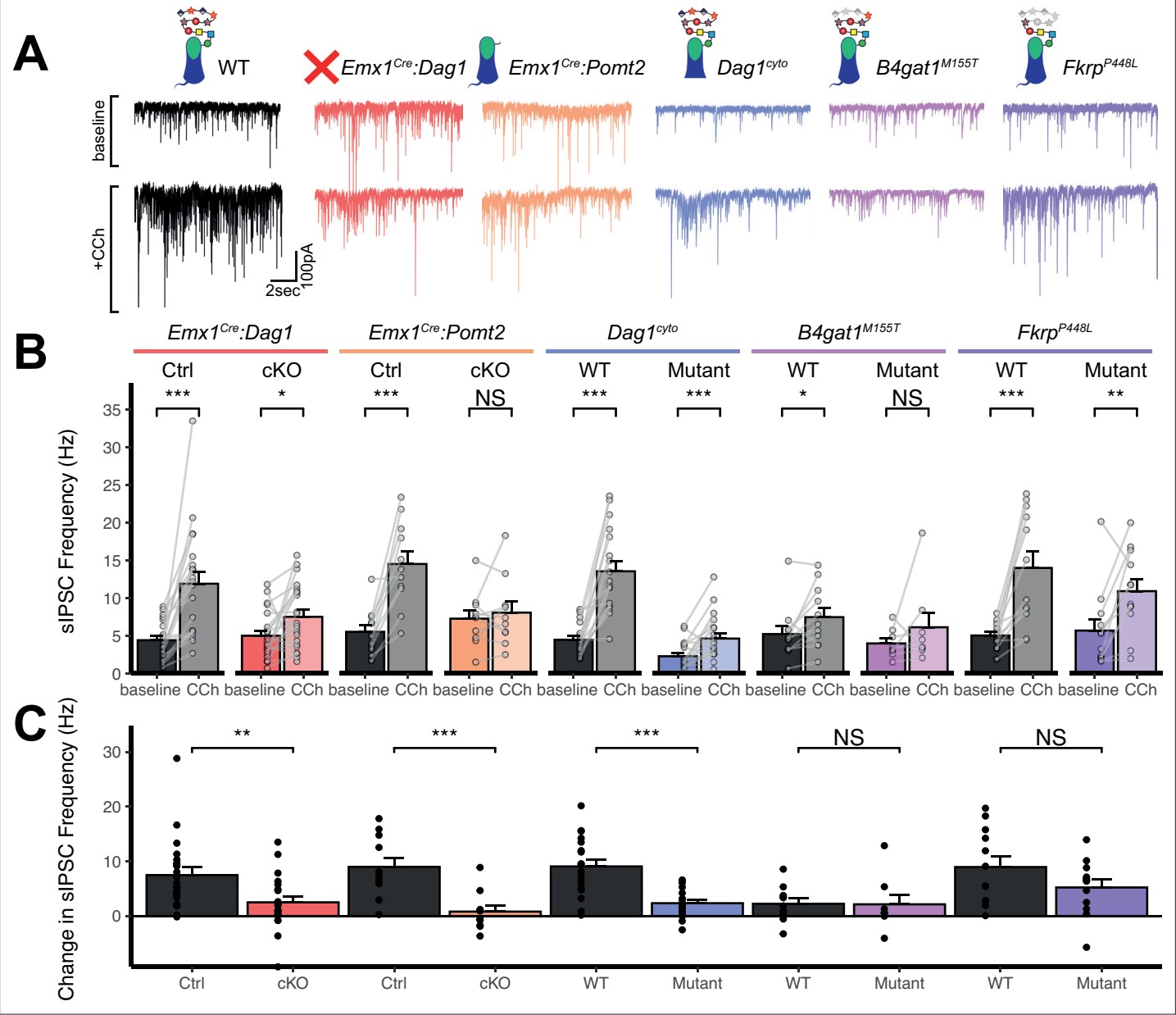

**Figure 6.** *Dag1* is required for CCK$^+$/CB$_1$R$^+$ IN synapse function in hippocampal CA1 in a manner dependent on both glycosylation and intracellular interactions. (**A**) Representative traces showing + seconds of sIPSC recordings at baseline (top) and after the addition of carbachol (bottom). (**B**) Quantification of average sIPSC frequency at baseline and after the addition of carbachol. (**C**) Quantification of the change in sIPSC frequency with the addition of carbachol. Error bars show mean + SEM. See *Supplementary file 1* for Ns. *Significance: *=p < 0.05, **=p < 0.01, ***=p < 0.001, NS = p ≥ 0.05. Abbreviations: sIPSC, spontaneous inhibitory postsynaptic current; CCh, carbachol.*

The online version of this article includes the following source data and figure supplement(s) for figure 6:

**Source data 1.** Raw data for quantification in *Figure 6B-C* and *Figure 6—figure supplement 1A, B*.

**Figure supplement 1.** Additional quantification of sIPSC recordings.

Dystroglycan may play a critical role in mediating the assembly of functional postsynaptic signaling/receptor complexes at these synapses.

Neither of the more mildly hypoglycosylated mutants (*B4gat1*$^{M155T/M155T}$, *Fkrp*$^{P448L/P448L}$) were different from their respective littermate controls in terms of the magnitude of the CCh effect on sIPSC frequency (*Figure 6C*), although the *B4gat1*$^{WT}$ mice appeared to possess a reduced effect of CCh compared to other control conditions (*Figure 6A–C*). The *B4gat1* line is of a mixed genetic background, which

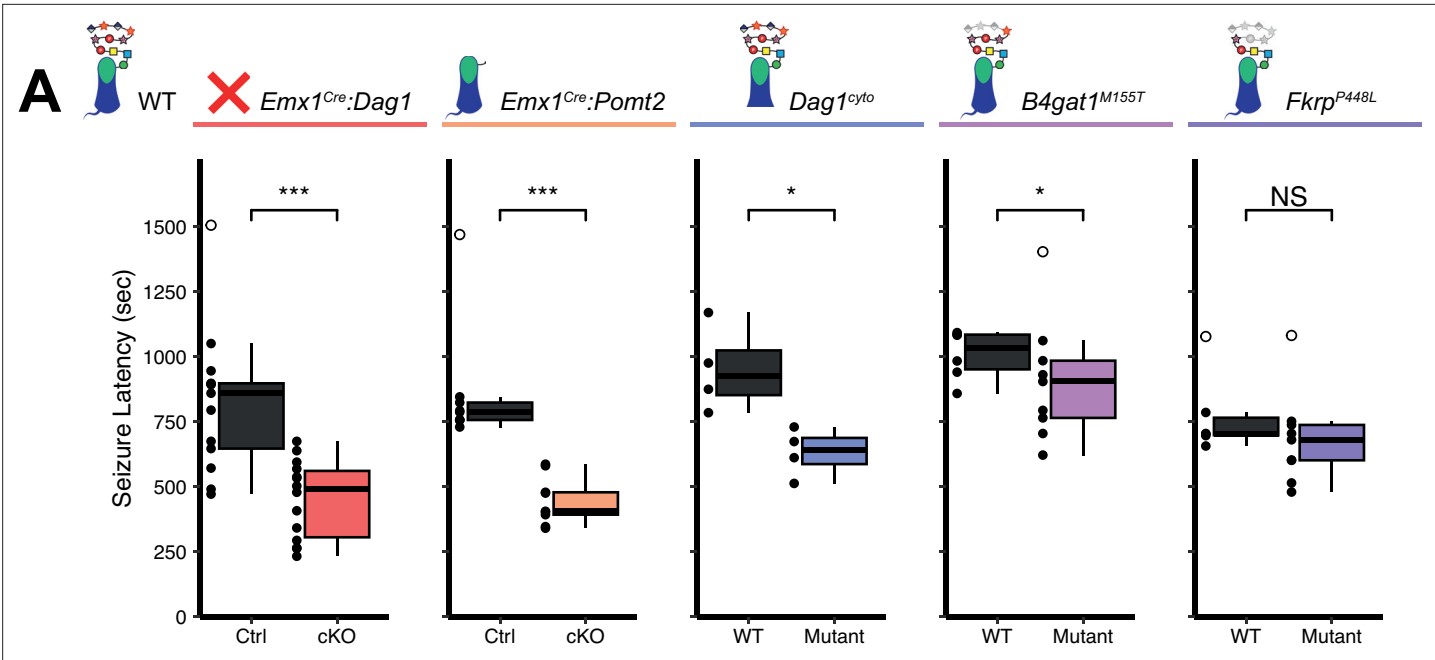

**Figure 7.** Reduced seizure induction threshold in models of dystroglycanopathy. (**A**) Quantification of latency (in seconds) to generalized tonic clonic seizure upon exposure to 10% flurothyl delivered at a constant rate. Open points denote statistical outliers. See **Supplementary file 1** for Ns. *Significance: \*=p < 0.05, \*\*=p < 0.01, \*\*\*=p < 0.001, NS = p ≥ 0.05.*

The online version of this article includes the following source data and figure supplement(s) for figure 7:

**Source data 1.** Raw data for quantification in **Figure 7A** and **Figure 7—figure supplement 1A–C**.

**Figure supplement 1.** Extended seizure induction threshold data.

could possibly explain the difference in CCh response. This finding is of unclear significance and may have obscured potential differences. Importantly, however, the marked functional synaptic differences observed between the *Emx1$^{Cre}$:Pomt2* cKO, *Emx1$^{Cre}$:Dag1* cKO and *Dag1$^{cyto/-}$* mice when compared with each of their respective controls described above was not seen in either of these phenotypically milder mutants.

Together, these results suggest that Dystroglycan is required for the function of CCK$^+$/CB$_1$R$^+$ IN perisomatic basket synapses in a glycosylation-dependent manner, as evidenced by the *Emx1$^{Cre}$:Dag1* cKO and *Emx1$^{Cre}$:Pomt2* cKO synaptic phenotypes, and that the intracellular domain of Dystroglycan is also required for normal CCK$^+$/CB$_1$R$^+$ IN basket synapse function. However, we cannot rule out the possibility that CCK$^+$/CB$_1$R$^+$ INs are simply less responsive to CCh in the mutants, as we lack the tools to identify CCK$^+$/CB1R$^+$ INs in live tissue for targeted recordings. In contrast, *B4gat1$^{M155T/M155T}$* and *Fkrp$^{P448L/P448L}$* hypomorphic mutants both appear to retain sufficient Dystroglycan glycosylation to maintain normal synapse function.

## Increased seizure susceptibility in models of dystroglycanopathy

Human patients with dystroglycanopathy have an increased risk of seizures and epilepsy (*Al Dhaibani et al., 2018*; *Di Rosa et al., 2011*; *Raphael et al., 2014*; *Yang et al., 2022*), however the underlying cause has yet to be determined. The observed defects in inhibitory basket synapse function suggest that alterations in neuronal circuit inhibition could potentially predispose mutant mice to seizures. To test whether mouse models of dystroglycanopathy exhibit a reduced seizure threshold, we exposed mice to the volatile chemoconvulsant flurothyl and measured the latency to generalized tonic-clonic seizure (TCS) (*Egawa et al., 2021*).

The latency to TCS was significantly faster in *Emx1$^{Cre}$:Dag1* cKO mice than their littermate controls (a 40.9% reduction on average, *Figure 7A*), with no difference in seizure latency between sexes in either group (*Figure 7—figure supplement 1C*). *Emx1$^{Cre}$:Pomt2* cKOs and *Dag1$^{cyto/-}$* mutants also had a significantly shorter latency to TCS than littermate controls (42.9% and 33.6% reductions,

respectively; *Figure 7A*), indicating that the mechanism underlying Dystroglycan's role in seizure susceptibility requires both extracellular glycosylation and intracellular interactions. *B4gat1^M155T/M155T* mutants showed a small but significant reduction (16%) in seizure latency, despite exhibiting no detectable functional deficit by electrophysiology (*Figure 6A–C*, *Figure 7A*). Finally, *Fkrp^P448L/P448L* mutants showed no significant change in seizure susceptibility (*Figure 7A*). Thus, the reduction in seizure latency reflects the severity of the synaptic phenotypes across the various models of dystroglycanopathy. These results demonstrate that disruptions in Dystroglycan function, including both its extracellular glycosylation and intracellular interactions, increase sensitivity to seizures.

## Discussion

Recent work identified a key role for neuronal Dystroglycan in the establishment and function of CCK^+/CB_1R^+ inhibitory synapses in the forebrain (*Früh et al., 2016*; *Miller and Wright, 2021*). Deletion of *Dag1* selectively from pyramidal neurons (*NeuroD6^Cre:Dag1* cKO) led to a near complete loss of CCK^+/CB_1R^+ INs during the first few postnatal weeks. In this study, we sought to better understand how CCK^+/CB_1R^+ IN synapse formation is affected in mouse models that more accurately reflect dystroglycanopathy, in which Dystroglycan function is more broadly affected throughout the CNS (*Figures 1 and 2*). Using a model that deletes *Dag1* throughout the developing forebrain (*Emx1^Cre:Dag1* cKO) we found that CCK^+/CB_1R^+ INs were present, but the laminar organization of their axon terminals and their ability to form functional basket synapses onto pyramidal neuron cell bodies in the hippocampus was impaired (*Figures 3–6*). The inability of CCK^+/CB_1R^+axon terminals to concentrate in the CA1-3 cell body layer began to manifest during the first postnatal week, when dynamic changes in laminar innervation by CCK^+/CB_1R^+ axons normally occur (*Figure 4*). Furthermore, these mice were found to exhibit a reduced seizure threshold compared to controls, showing for the first time that mouse models of dystroglycanopathy are vulnerable to seizures (*Figure 7*). Because *Emx1^Cre* (and *Nestin^Cre*) conditional deletion of *Dag1* or *Pomt2* leads to widespread loss of functional Dystroglycan in the forebrain in contrast with the previously studied *NeuroD6^Cre* conditional deletion, which targets pyramidal neurons, these models more accurately model dystroglycanopathy.

We found that CCK^+/CB_1R^+ IN synapse formation and function are dependent on proper Dystroglycan glycosylation and appear to correlate with the degree of hypoglycosylation in different mutants. Complete reduction of glycosylation in *Emx1^Cre:Pomt2* cKO mutants caused the same phenotypes seen in Dystroglycan conditional knockouts (*Emx1^Cre:Dag1* cKO; *Figures 1–3 and 5–6*), possibly due to the mislocalization of Dystroglycan. The finding that glycosylation is required for Dystroglycan synaptic localization in hippocampal pyramidal cells is similar to a previous finding in retinal photoreceptors in the context of *Pomt1* conditional deletion (*Rubio-Fernández et al., 2018*). In contrast, when *Fktn* deletion is induced in myotubes β-Dystroglycan localization is unchanged, suggesting that this phenomenon is unique to synaptic Dystroglycan (*Beedle et al., 2012*). One interpretation is that without matriglycan present to mediate interaction with presynaptic cells, Dystroglycan is no longer concentrated at synaptic sites, implicating it as a synaptic organizer. However, the miswiring of the CCK^+/CB_1R^+ axons could also reduce the likelihood of postsynaptic Dystroglycan encountering a presynaptic axon, discouraging synaptic localization. Conversely, it is possible that glycosylation is required for trafficking to the surface in the first place, however this is less likely given that the levels of β-Dystroglycan were normal in membrane-enriched lysate (*Figure 1*, *Figure 1—figure supplement 2*).

A milder reduction in glycosylation (*B4gat1^M155T/M155T*) resulted in a cortical migration phenotype that was restricted to midline (*Figure 3—figure supplement 4*) and a small reduction in CCK^+/CB_1R^+ axon terminals and synaptic puncta density in CA1 which did not appear to affect synapse function (*Figures 3 and 5–6*). The mildest reduction in glycosylation amongst our models was observed in *Fkrp^P448L/P448L* mutants, which exhibited normal cortical migration but the same mild defect in CCK^+/CB_1R^+ IN axon targeting and synaptic puncta density observed in *B4gat1^M155T/M155T* mutants (*Figures 3 and 5–6*). Together, these three glycosylation mutants illustrate the degree of hypoglycosylation required for neurodevelopmental processes and show that defects in synaptic function only arise in the context of severely reduced glycosylation; the residual Dystroglycan function present in *B4gat1^M155T/M155T* and *Fkrp^P448L/P448L* mutants is sufficient for most aspects of brain development. Finally, using *Dag1^cyto/-* mutants that lack the intracellular domain of Dystroglycan, we found that the intracellular domain plays a role in some, but not all, neurodevelopmental processes. The intracellular domain

is not required for neuronal migration in neocortex or synapse formation in CA1 (*Figures 2 and 5*) but is required for the proper targeting of CCK⁺/CB₁R⁺ IN axons in CA1-3 (*Figure 3*) and for subsequent CCK⁺/CB₁R⁺ IN basket synapse function (*Figure 6*).

## Dystroglycan is an essential transsynaptic organizing molecule for CCK⁺/CB₁R⁺ basket synapses

Synaptogenesis requires multiple distinct steps: (1) synaptic partner recognition, (2) recruitment and assembly of core pre- and post-synaptic machinery, (3) differentiation and maturation of synaptic identity, and (4) synaptic maintenance (*Südhof, 2018*). Based on data from this study (*Figure 4*) and previous work from our group and others, mice lacking *Dystroglycan* exhibit defects in CCK⁺/CB₁R⁺ IN development at the earliest time point they can be reliably identified (P0-P5), before the peak phase of inhibitory synapse formation (P9), suggesting that Dystroglycan functions at the earliest stages of synaptogenesis such as synaptic partner recognition (*Favuzzi et al., 2019*). Determining the precise onset of synapse targeting and formation for most IN subtypes, including CCK⁺/CB₁R⁺INs, is limited by a lack of genetic tools for visualizing and manipulating IN subtypes during developmental stages.

The impairment in CCK⁺/CB₁R⁺ IN development throughout the forebrain suggests a trans-synaptic role for Dystroglycan (*Figure 3*, *Figure 3—figure supplement 4*). The identity of the trans-synaptic binding partner between Dystroglycan-expressing cells and CCK⁺/CB₁R⁺ INs remains unknown. Our data in *Emx1^{Cre}:Pomt2* cKO mice point to a critical role for the glycan chains on Dystroglycan mediating this binding. All proteins that bind to the glycan chains on Dystroglycan do so through at least one Laminin G (LG) domain. There are over 25 LG-domain containing extracellular or transmembrane proteins expressed in the hippocampus. Neurexins, a family of highly alternatively spliced synaptic cell-adhesion molecules (*NRXN1-3*) which each contain multiple LG domains, bind Dystroglycan in a glycosylation-dependent manner (*Boucard et al., 2005*; *Fuccillo et al., 2015*; *Reissner et al., 2014*; *Sugita et al., 2001*). The specific splice isoforms of *Nrxns* that bind Dystroglycan are expressed by CCK⁺/CB₁R⁺ INs (*Fuccillo et al., 2015*; *Ullrich et al., 1995*). *Neurexin-3* conditional knockout (targeting all *Nrxn3* isoforms) and CRISPR-mediated *Dag1* knockout both result in similar synaptic deficits in olfactory bulb and prefrontal cortex (*Trotter et al., 2023*). While a Dystroglycan knock-in mouse with reduced glycosylation that impairs Neurexin binding (*Dag1^{T190M}*) shows normal CCK⁺/CB₁R⁺ synapse formation by immunohistochemistry, the functionality of these synapses was not assessed by electrophysiology (*Früh et al., 2016*). Similar to *B4gat1^{M155T/M155T}* and *Fkrp^{P448L/P448L}* mutants, the *Dag1^{T190M}* mutation does not fully eliminate Dystroglycan glycosylation, and therefore does not rule out the possibility that Neurexins play a role at CCK⁺/CB₁R⁺ synapses. It is also possible that a yet undescribed Dystroglycan interacting protein is required for initial synapse recognition, and Nrxn-Dag1 interactions are required for subsequent synapse maturation and maintenance only. Indeed, the majority of studies indicate that Neurexins are not required for the initial formation of synapses, but rather regulate the maturation and structural maintenance of synapses after they have formed (*Chen et al., 2017*; *Dudanova et al., 2007*; *Lin et al., 2023*; *Missler et al., 2003*; *Trotter et al., 2023*). Interestingly, while Dystroglycan localizes to both PV⁺ and CCK⁺/CB₁R⁺ inhibitory basket synapses in CA1, only the CCK⁺/CB₁R⁺ IN population was affected in the dystroglycanopathy models (*Früh et al., 2016*). Presumably, PV⁺ INs have a distinct developmental program independent of Dystroglycan and likely require a different postsynaptic recognition partner.

## A role for the dystrophin-glycoprotein complex in CCK⁺/CB₁R⁺ interneuron development

In brain and muscle tissue, Dystroglycan forms a complex with Dystrophin and several other proteins, collectively known as the Dystrophin Glycoprotein Complex (DGC). Like Dystroglycan, Dystrophin is also expressed throughout the forebrain and is associated with inhibitory synapses in multiple brain regions (*Knuesel et al., 1999*). Patients with mutations in *Dystrophin* develop Duchenne Muscular Dystrophy (DMD), and frequently exhibit cognitive impairments in the absence of brain malformations, suggesting a general role for the DGC in synapse development and function (*Jagadha and Becker, 1988*; *Moizard et al., 2000*; *Naidoo and Anthony, 2020*). A mouse model of DMD lacking all neuronal Dystrophin isoforms (*mdx*) exhibits defects in CCK⁺/CB₁R⁺ IN synapse development and abnormal innervation in the hippocampus, resembling the innervation pattern we observed in *Emx1^{Cre}:Dag1* cKO and *Emx1^{Cre}:Pomt2* cKO mice in this study (*Krasowska et al., 2014*). Since Dystroglycan interacts

with Dystrophin through its intracellular domain, we expected to observe similar phenotypes in mice lacking the intracellular domain of Dystroglycan (*Dag1^cyto/-*). However, *Dag1^cyto/-* showed a milder axon targeting defect than *Emx1^Cre:Dag1* cKO or *mdx* mice. In addition, IIH6 puncta were normally localized to the somatodendritic compartment in *Dag1^cyto/-* mutants, suggesting that Dystroglycan does not require interactions with Dystrophin for is localization to somatodendritic synapses. However, Dystroglycan's synaptic localization has not been examined in the *mdx* mutants. Clearly, additional work is required to better understand the relationship between Dystroglycan and Dystrophin at synapses in the brain.

While the density of CCK^+/CB_1R^+ IN synaptic puncta was normal in *Dag1^cyto/-* mice, synaptic function was impaired to the same level as *Emx1^Cre:Dag1* cKO and *Emx1^Cre:Pomt2* cKO mice, and seizure latency was reduced. Given that *Dag1^cyto/-* and *B4gat1^M155T/M155T* mutants show a similar reduction in Dystroglycan glycosylation (*Figure 1B–C*), our observation that the functional synaptic phenotype is restricted to the *Dag1^cyto/-* mutant reinforces the notion that the intracellular domain of Dystroglycan plays an active role in organizing essential postsynaptic signaling elements. One possibility is that the intracellular domain of Dystroglycan is required to recruit additional postsynaptic scaffolding elements and receptors necessary for CCK^+/CB_1R^+ basket synapse function (*Uezu et al., 2019*). Importantly, *Dag1^cyto/-* mice did not show a cortical migration phenotype (*Figure 2A–D*), indicating that the functional synaptic deficits and reduced seizure latency occurred independent of cortical malformation.

## Altered inhibitory synapse development and function may contribute to neurological symptoms in dystroglycanopathy

In addition to muscular atrophy and hypotonia, dystroglycanopathy patients often present with central nervous system symptoms. Patients with the most severe forms of dystroglycanopathy (FCMD, Muscle-Eye-Brain disease, and Walker-Warburg Syndrome) exhibit structural changes including hypoplasia of the retina, brainstem and spinal cord, cerebellar cysts, hydrocephalus, Type II lissencephaly, and microcephaly, associated with seizures and cognitive disability (*Meilleur et al., 2014*; *Mercuri et al., 2009*). Patients with milder forms of dystroglycanopathy may show cognitive disability and/or seizures without gross brain malformations, suggesting that there may be synaptic deficits independent of early neurodevelopmental processes (e.g. neuronal migration, axon guidance; *Mercuri et al., 2009*; *Yang et al., 2022*). The mouse models used in this study recapitulate the full spectrum of brain malformations seen in human patients. *Emx1^Cre:Dag1* cKO and *Emx1^Cre:Pomt2* cKO mice show Type II lissencephaly consistent with severe dystroglycanopathy, whereas *B4gat1^M155T/M155T* and *Fkrp^P448L/P448L* mutants have relatively normal cortical development consistent with mild dystroglycanopathy. Mutations in any of the genes involved in the glycosylation of Dystroglycan can result in dystroglycanopathy with seizures, but the incidence and severity of seizures is higher in patients with brain malformations (*Mercuri et al., 2009*; *Wang et al., 2017*; *Yang et al., 2022*).

Mice have been used to model dystroglycanopathy for decades; however, to our knowledge the present study is the first to investigate seizure susceptibility in mouse models of dystroglycanopathy. It is probable that the CCK^+/CB_1R^+ interneuron axon targeting and synapse phenotypes in the mouse models described in the present study contribute to their seizure susceptibility and open the possibility that defective inhibitory synaptic signaling mechanisms may underlie seizures in dystroglycanopathy patients. Although severe neuronal migration phenotypes In *Emx1^Cre:Dag1* cKO and *Emx1^Cre:Pomt2* cKO mice may contribute to seizure activity, our observation that *Dag1^cyto/-* mutants showed both abnormal CCK^+/CB_1R^+synaptic function and reduced seizure latency, with intact cortical migration, indicates that the seizure phenotype is likely associated with synaptic defects. Supporting these results, CCK^+/CB_1R^+ interneurons in the hippocampus are selectively lost in models of temporal lobe epilepsy with recurrent seizures induced by pilocarpine. CCK^+/CB_1R^+ axons in CA1-3 begin to degenerate within hours of status epilepticus, whereas PV^+ INs are unaffected in this model (*Whitebirch et al., 2023*; *Wyeth et al., 2010*).

While our *B4gat1^M155T/M155T* mutants showed only a slightly reduced seizure latency, the mutants experienced more severe seizures than the other mouse models, resulting in death in 50% of cases (4/8 mutants compared to 0/6 fatalities among littermate controls) (*Figure 7—figure supplement 1A*). Flurothyl-induced seizures are typically generalized forebrain seizures; however in seizure-prone mouse models or in mice exposed to higher concentrations of flurothyl, mice can experience a suppression of brainstem oscillations followed by sudden death (*Gu et al., 2022*; *Kadiyala et al., 2016*). The

$B4gat1^{M155T/M155T}$ mutation was originally identified based on a hindbrain axon guidance phenotype, suggesting they may have currently unknown defects in brainstem development or circuitry that could render them more susceptible to fatal brainstem seizures (*Wright et al., 2012*). Because the *Dag1* and *Pomt2* mutants are forebrain-specific conditional knockouts, (*Figure 1—figure supplement 1A*), we would not anticipate abnormal axon guidance in the brainstem or hindbrain of these mutants. Further research on the nature and progression of seizures observed in mouse models may have a profound impact on our understanding of dystroglycanopathy and potential therapeutic interventions.

### Potential therapeutics for the restoration of synapse function in patients with dystroglycanopathy

Most patients with dystroglycanopathy present with mutations in one of the 19 genes required for the glycosylation of Dystroglycan, resulting in hypoglycosylated Dystroglycan. We have demonstrated that a mild reduction in the glycosylation of Dystroglycan, as seen in $Fkrp^{P448L/P448L}$ and $B4gat1^{M155T/M155T}$ mutants, does not significantly disrupt synapse function. This suggests that glycosylation may not need to be restored to wild-type levels in order to achieve normal synapse function. Gene replacement therapy may be well suited to treat certain forms of dystroglycanopathy by rescuing glycosylation. AAV-mediated delivery of fully functional glycosyltransferases has been shown to significantly improve muscle pathology and function in dystrophic mice, however synaptic phenotypes have not been examined (*Kanagawa, 2021*). Supplementation with (CDP)-ribitol, which is synthesized by *Crppa* (previously known as *ISPD*), can restore functional Dystroglycan glycosylation and improve muscle function in mouse models with hypomorphic mutations in *Crppa* or *Fkrp* (*Cataldi et al., 2018*). In mice lacking functional *Crppa* or *Fkrp* in skeletal muscle, (CDP)-ribitol can further enhance the therapeutic impact of gene restoration (*Cataldi et al., 2020*). However, whether (CDP)-ribitol treatment can improve Dystroglycan function in other models of dystroglycanopathy, or is capable of restoring Dystroglycan glycosylation and synaptic function in the nervous system, remains untested.

### Conclusions

We demonstrate that Dystroglycan is critical for the postnatal development of $CCK^+/CB_1R^+$ interneuron axon targeting and synapse formation/function in the hippocampus of severe mouse models of dystroglycanopathy. Extracellular glycosylation of Dystroglycan and intracellular interactions involving the cytoplasmic domain are both essential for Dystroglycan's synaptic organizing role. Mice with a partial reduction in glycosylation have relatively normal $CCK^+/CB_1R^+$ interneuron axon targeting and synapse function, suggesting that even a partial restoration of glycosylation may have some therapeutic benefit. These findings suggest that $CCK^+/CB_1R^+$ interneuron axon targeting defects may contribute to cognitive impairments and seizure susceptibility in dystroglycanopathy.

**Table 1.** Mouse strains.

| Common name | Strain name | Reference | Stock # |
|---|---|---|---|
| $Dag1^{Flox}$ | $B6.129(Cg)-Dag1^{tm2.1Kcam}/J$ | *Cohn et al., 2002* | 009652 |
| $Dag1^{cyto}$ | N/A | *Satz et al., 2009* | N/A |
| $Pomt2^{Flox}$ | $POMT2tm1.1Hhu/J$ | *Hu et al., 2011* | 017880 |
| $B4gat1^{M155T}$ | $B6(C3)-B4GAT1^{m1Ddg}/J$ | *Wright et al., 2012* | 022018 |
| $Fkrp^{P448L}$ | $C57BL/6NJ-Fkrp^{em1Lgmd}/J$ | *Chan et al., 2010* | 034659 |
| $R26^{LSL-H2B-mCherry}$ | $B6.Gt(ROSA)26Sor^{tm1.1Ksvo}$ | *Peron et al., 2015* | 023139 |
| $Emx1^{Cre}$ | $B6.129S2-Emx1^{tm1(cre)Krj}/J$ | *Gorski et al., 2002* | 005628 |
| $Nestin^{Cre}$ | $B6.Cg-Tg(Nes-cre)^{1Kln}/J$ | *Tronche et al., 1999* | 003771 |
| $NEX^{Cre}$ | $NeuroD6^{tm1(cre)Kan}$ | *Goebbels et al., 2006* | MGI:4429523 |
| $Sox2^{Cre}$ | $B6N.Cg-Edil3^{Tg(Sox2-cre)1Amc}/J$ | *Hayashi et al., 2002* | 014094 |

**Table 2.** Breeding schemes.

| Breeding Scheme | Control Genotype | Mutant Genotype |
|---|---|---|
| $Emx1^{Cre/+};Dag1^{+/-}$ x $Dag1^{Flox/Flox}$ | $Emx1^{Cre/+};Dag1^{Flox/+}$ | $Emx1^{Cre/+};Dag1^{Flox/-}$ |
| $Nestin^{Cre/+};Dag1^{+/-}$ x $Dag1^{Flox/Flox}$ | $Nestin^{Cre/+};Dag1^{Flox/+}$ | $Nestin^{Cre/+};Dag1^{Flox/-}$ |
| $NEX^{Cre/+};Dag1^{+/-}$ x $Dag1^{Flox/Flox}$ | $NEX^{Cre/+};Dag1^{Flox/+}$ | $NEX^{Cre/+};Dag1^{Flox/-}$ |
| $Emx1^{Cre/+};Pomt2^{Flox/+}$ x $Pomt2^{Flox/Flox}$ | $Emx1^{Cre/+};Pomt2^{Flox/+}$ | $Emx1^{Cre/+};Pomt2^{Flox/Flox}$ |
| $Dag1^{cyto/+}$ x $Dag1^{+/-}$ | WT | $Dag1^{cyto/-}$ |
| $B4gat1^{M155T/+}$ x $B4gat1^{M155T/+}$ | WT | $B4gat1^{M155T/M155T}$ |
| $Fkrp^{P448L/+}$ x $Fkrp^{P448L/+}$ | WT | $Fkrp^{P448L/P448L}$ |

# Materials and methods

## Animal husbandry

All animals were housed and cared for by the Department of Comparative Medicine (DCM) at Oregon Health and Science University (OHSU), an AAALAC-accredited institution. Animal procedures were approved by OHSU Institutional Animal Care and Use Committee (Protocol # IS00000539), adhered to the NIH *Guide for the care and use of laboratory animals*, and provided with 24 hr veterinary care. Animal facilities are regulated for temperature and humidity and maintained on a 12 hr light-dark cycle and were provided food and water ad libitum. Animals older than postnatal day 6 (P6) were euthanized by administration of $CO_2$, animals <P6 were euthanized by rapid decapitation.

## Mouse strains and genotyping

The day of birth was designated postnatal day 0 (P0). Ages of mice used for each analysis are indicated in the figure and figure legends. Mouse strains used in this study have been previously described and were obtained from Jackson Labs, unless otherwise indicated (*Table 1*; *Chan et al., 2010*; *Cohn et al., 2002*; *Goebbels et al., 2006*; *Gorski et al., 2002*; *Hu et al., 2011*; *Peron et al., 2015*; *Satz et al., 2009*; *Tronche et al., 1999*; *Wright et al., 2012*). $Dag1^{+/-}$ mice were generated by crossing the $Dag1^{flox/flox}$ line to a $Sox2^{Cre}$ line to generate germline $Dag1^{\Delta/+}$ mice hereafter referred to as $Dag1^{+/-}$ as the resultant transcript is nonfunctional. These mice were thereafter maintained as heterozygotes. Breeding schemas are as described in *Table 2*. Where possible, mice were maintained on a C57BL/6J background. $Dag1^{cyto/-}$ mice occurred at a frequency lower than Mendelian, suggesting that a proportion of progeny die embryonically. To increase viability of pups, the $Dag1^{cyto}$ line was outcrossed to a CD-1 background for one generation. The $B4gat1$ line has a mixed genetic background: it was founded on a C3H/He background and then crossed on to C57BL/6J for future generations. Genomic DNA extracted from toe or tail samples (Quanta BioSciences) was used to genotype animals. Primers for genotyping can be found on the JAX webpage or originating article. $Dag1^{+/-}$ mice were genotyped with the following primers: CGAACACTGAGTTCATCC (forward) and CAACTGCTGCATCTCTAC (reverse). For each mouse strain, littermate controls were used for comparison with mutant mice. For all experiments, unless otherwise noted, mice of both sexes were used indiscriminately. See *Supplementary file 1* for a summary of sexes used in each experiment.

## Perfusions and tissue preparation

Brains from mice younger than P15 were dissected and drop fixed in 5 mLs of 4% paraformaldehyde (PFA) in phosphate buffered saline (PBS) overnight for 18–24 hr at 4 °C. Mice P15 and older were deeply anesthetized using CO2 and transcardially perfused with ice cold 0.1 M PBS for two minutes to clear blood from the brain, followed by 15 mL of ice cold 4% PFA in PBS. After perfusion, brains were dissected and post-fixed in 4% PFA for 30 min at room temperature. Brains were rinsed with PBS, embedded in 4% low-melt agarose (Fisher cat. no. 16520100), and sectioned at 50 µm using a vibratome (VT1200S, Leica Microsystems Inc, Buffalo Grove, IL) into 24-well plates containing 1 mL of 0.1 M PBS with Sodium Azide.

**Table 3.** Primary antibodies used for immunohistochemistry.

| Target | Host species | Dilution | Source | Catalog # | RRID |
|---|---|---|---|---|---|
| α-Dystroglycan (IIH6C4) | Mouse | 1:250 | Millipore | 05–593 | AB_309828 |
| β-Dystroglycan | Mouse | 1:50 | Leica Biosystems | NCL-b-DG | AB_442043 |
| CB1R | Guinea pig | 1:1000 | Synaptic Systems | 258–104 | AB_2661870 |
| Cux1 | Rabbit | 1:500 | Santa Cruz Biotech | sc-13024 | AB_2261231 |
| Laminin | Rabbit | 1:1000 | Sigma-Aldrich | L9393 | AB_477163 |
| NECAB1 | Rabbit | 1:500 | Sigma-Aldrich | HPA023629 | AB1848014 |
| NECAB2 | Rabbit | 1:500 | Proteintech | 12257–1-AP | AB_2877841 |
| NeuN | Mouse | 1:250 | Millipore | MAB377 | AB_2298772 |
| Parvalbumin | Rabbit | 1:1000 | Swant | PV27 | AB_2631173 |
| Parvalbumin | Mouse | 1:50 | Swant | 235 | AB_10000343 |
| Somatostatin | Rabbit | 1:2000 | Peninsula Labs | T-4103 | AB_518614 |
| Tbr1 | Rabbit | 1:500 | Millipore | AB10554 | AB_10806888 |
| VGAT | Rabbit | 1:500 | Synaptic Systems | 131–003 | AB_887869 |
| VGAT | Guinea Pig | 1:500 | Synaptic Systems | 131–005 | AB_1106810 |
| VGlut3 | Rabbit | 1:1000 | Synaptic Systems | 135–203 | AB_887886 |
| VIP | Rabbit | 1:1000 | ImmunoStar | 20077 | AB_572270 |

## Immunohistochemistry

Single and multiple immunofluorescence detection of antigens was performed as follows: free-floating vibratome sections (50 μm) were briefly rinsed with PBS, then blocked for 1 hr in PBS containing 0.2% Triton-X (PBST) plus 2% normal goat or donkey serum. Sections were incubated with primary antibodies (*Table 3*) diluted in blocking solution at 4 °C for 48–72 hr. For staining of Dystroglycan synaptic puncta, an antigen retrieval step was performed prior to incubation in primary antibody. Briefly, sections were incubated in sodium citrate solution for 15 min at 95°C in a water bath followed by 15 minutes at room temperature. Following incubation in primary antibody, sections were rinsed with PBS then washed with PBST three times for 20 min each. Sections were then incubated with a cocktail of secondary antibodies (1:500, Alexa Fluor 488, 546, 647) in blocking solution overnight at room temperature. Sections were washed with PBS three times for 20 min each and counterstained with Hoechst 33342 (1:10,000, Life Technologies, Cat# H3570) for 20 min to visualize nuclei. Finally, sections were mounted on slides using Fluoromount-G (SouthernBiotech) and sealed using nail polish.

## Microscopy

Imaging was performed on either a Zeiss Axio Imager M2 fluorescence upright microscope equipped with an Apotome.2 module or a Zeiss LSM 980 laser scanning confocal build around a motorized Zeiss Axio Observer Z1 inverted microscope with a Piezo stage. The Axio Imager M2 uses a metal halide light source (HXP 200 C), Axiocam 506 mono camera, and 10X/0.3 NA EC Plan-Neofluar, 20X/0.8 NA Plan-Apochromat objectives. The LSM 980 confocal light path has two multi-alkali PMTs and two GaAsP PMTs for four track imaging. Confocal images were acquired using a 63X/1.4 NA Plan-Apochromat Oil DIC M27 objective. Z-stack images were acquired and analyzed offline in ImageJ/FIJI (*Schindelin et al., 2012*) or Imaris 9.8 (Oxford Instruments). Images used for quantification between genotypes were acquired using the same exposure times. Brightness and contrast were adjusted in FIJI to improve visibility of images for publication. Figures were composed in Adobe Illustrator 2023 (Adobe Systems).

## Image quantification

For imaging experiments, 4–8 images were acquired from 2 to 4 coronal sections per animal, and at least three animals per genotype were used for analysis.

## Cortical lamination

Images of somatosensory cortex were acquired using a 10X objective on a Zeiss Axio Imager M2. 4 µm z-stacks covering 16 µm were acquired and multiple tiles were stitched together. Maximum projections were used for analysis. In FIJI, the straight line tool with a 300 µm line width was used to measure the fluorescence profile from corpus callosum to pial surface. Background fluorescence was determined as the average fluorescence of the 20 darkest pixels; background was then subtracted from all points. The cortical distance was broken into 10 bins and average fluorescence within each bin was compared between genotypes.

## Hippocampal CA1 CB$_1$R and PV distribution

Images of dorsal hippocampal CA1 were acquired using a 20X objective on a Zeiss Axio Imager M2. Maximum projection images of 0.6 µm z-stacks covering 9 µm were analyzed in FIJI. The straight line tool with a 300 µm line width was used to measure the fluorescence profile within SO, SP, and SR of CA1, avoiding Parvalbumin$^+$ cell bodies. Background fluorescence was determined as the average fluorescence of the 50 darkest pixels; background was then subtracted from all points. The thickness of SP was determined using Hoechst fluorescence. Average fluorescence within SO/SP/SR was compared between genotypes.

## Interneuron cell counts in CA1

Images of dorsal hippocampal CA1 were acquired using a 10X objective on a Zeiss Axio Imager M2. Maximum projection images of 4 µm z-stacks covering 40 µm were analyzed in FIJI. Immunolabeled NECAB1/NECAB2/PV cell bodies were counted if they were within 100 µm of *stratum pyramidale*. The freehand line tool was used to measure the length of *stratum pyramidale*. Cell number was normalized to the length of *stratum pyramidale* present in the analyzed region.

## Hippocampal CB$_1$R/PV/VGAT density and co-localization

Images of dorsal hippocampal CA1 were acquired using a 63X objective on a Zeiss LSM 980. 0.2 µm z-stacks covering 3 µm were analyzed in Imaris. Hoechst fluorescence was used to determine the bounds of SP. The Imaris Spots function was used to determine the location of synaptic puncta in 3-dimensional space. Synaptic puncta were deemed to be co-localized if they were within 1 µm of each other.

## Western blot

Cortex or hippocampus was dissected and solubilized in 1 mL of lysis buffer containing 100 mM NaCl, 50 nM Tris, 2.5 mM CaCl$_2$, 1% Triton X-100, 1% n-Octyl-β-D-glucopyranoside, and protease inhibitors. Lysate was incubated at 4 °C for 1 hr and then spun at 12,500 *g* for 25 min. Supernatant containing 3000 µg (cortex) or 2000 µg (hippocampus) of protein (as determined by Pierce BCA Protein Assay) was applied to agarose-bound Wheat Germ Agglutinin (WGA) (Vector Labs) overnight at 4 °C. Beads were washed 3 times in TBS and boiled in 1X LDS sample buffer with 2-Mercaptoethanol (1:100) for 5 min. Samples were run on a 4–15% gradient polyacrylamide gel at 100 V for 75 min and then transferred to a PVDF membrane (100 V for 100 min). For immunoblotting, membranes were blocked in 5% milk TBST and then incubated overnight at 4 °C in 5% milk TBST containing primary antibody. Antibodies used: α-Dystroglycan (IIH6C4) (Millipore cat. no. 05–593, RRID: AB_309828, mouse IgM, 1:500), MANDAG2 (DSHB cat. no. 7D11, RRID: AB_2211772, mouse IgG1, 1:500), β3-tubulin (Cell Signaling Technology cat. no. 5568, RRID: AB_10694505, rabbit, 1:2000). Membranes were washed 3 times in TBST for 15 minutes each and incubated on fluorescent IRDye secondary antibody (1:10,000, LI-COR) in 5% milk TBST for 1 hr at room temperature. Membranes were imaged using a LI-COR Odyssey CLx 0918 imager and signal analyzed using LI-COR Image Studio Lite version 5.2.

## Electrophysiology

For acute slice preparation, mice were deeply anesthetized in 4% isoflurane and subsequently injected with a lethal dose of 2% 2, 2, 2-Tribromoethanol in sterile water followed by transcardial perfusion with 10 mL ice cold cutting solution containing the following (in mM): 93 NMDG, 2.5 KCl, 1.2 NaH$_2$PO$_4$, 30 NaHCO$_3$, 20 HEPES, 24 glucose, 5 Na Ascorbate, 2 Thiourea, 3 Na Pyruvate, 13 N-Acetyl Cysteine, 1

Kynurenic acid, 10 MgSO₄, 0.5 CaCl₂; pH 7.3, 300–340 mmol/kg. After rapid decapitation, the brain was briefly submerged in ice cold cut solution bubbled with carbogen (95% oxygen, 5% CO$_2$) and then sectioned into 300 µm sagittal sections (Leica VT1200S vibratome) in bubbled ice-cold cut solution. Slices were recovered in 37 °C cut solution, bubbled, for 15 min followed by 1 hr in room temperature recording ACSF (containing, in mM: 125 NaCl, 25 NaHCO$_3$, 1.25 NaH$_2$PO$_4$, 3 KCl, 25 D-Glucose, 2 CaCl$_2$, 1 MgCl$_2$) with an osmolarity of 310–320 mmol/kg and supplemented with 1.5 mM Na Ascorbate, bubbled.

CA1 pyramidal cells were patched in whole cell configuration using 3–5 MΩ borosilicate glass pipettes filled with high chloride internal solution containing the following (in mM): 125 CsCl, 2.5 MgCl$_2$, 0.5 EGTA, 10 HEPES, 2 Mg-ATP, 0.3 Na-GTP, 5 QX-314; pH 7.2, 300 mmol/kg. Pipettes were wrapped in parafilm to reduce capacitive currents. Cells were voltage clamped at –70 mV and continuously superfused with 2–3 mL/min bubbled recording ACSF (310–320 mmol/kg) containing 10 µM NBQX to block excitatory transmission. Recordings were performed at 34 °C. After 5 min of spontaneous IPSC (sIPSC) recording, 10 µM Carbachol was added to the perfusate and another 5 min of sIPSC were recorded. Slices were discarded after exposure to Carbachol. Signals were amplified with an AxoPatch 200B amplifier (Molecular Devices), low-pass filtered at 5 kHz, and digitized and sampled at 10 kHz with a NIDAQ analog-to-digital board (National Instruments). Data were acquired and analyzed using a custom script in Igor Pro 8 (Wavemetrics; https://github.com/jnjahncke/mini_analysis copy archived at *Jahncke, 2023*). A hyperpolarizing step of –10 mV was applied before each sweep to monitor input resistance, series resistance, and measure cell capacitance. Series resistance was not compensated and was maintained below 20 MΩ. Cells were excluded if series resistance changed by more than 25%.

To calculate the proportion of cells that responded to Carbachol, cells were sorted into 'responsive' and 'non-responsive' categories. Cells were categorized as responsive if sIPSC frequency increased by 20% or more with the addition of Carbachol. If sIPSC frequency in a cell changed by less than 20% or less than 0.5 Hz it was deemed non-responsive.

## Flurothyl seizure induction

Mice aged P40-P55 were used for the flurothyl-induced seizure susceptibility assay to determine seizure threshold. Briefly, mice were placed in an enclosed glass chamber equipped with a vaporization chamber out of reach of the mouse. Volatile liquid 10% Bis (2,2,2-Trifluorotheyl) Ether (Millipore Sigma cat. no 287571) in 95% EtOH was delivered to the vaporization chamber at a rate of 6 mL/hr. Seizure latency was determined as the amount of time until generalized tonic-clonic seizure (TCS). Upon exhibiting TCS, animals were immediately removed from the chamber and returned to their home cage, whereupon seizures ceased rapidly. Sample size was determined using power analysis as described below. The *Emx1*$^{Cre}$:*Dag1* experimental groups were powered sufficiently to determine sex differences. Because no sex difference was found (*Figure 7—figure supplement 1C*), sexes were pooled for the remaining experiments. For statistical tests, outliers were excluded. Outliers were calculated as follows: first, the interquartile range (IQR) was calculated and multiplied by 1.5. This 1.5*IQR value was subtracted from the 25% quartile (Q1) and added to the 75% quartile (Q3). Points outside of the range Q1 - 1.5*IQR <x < Q3+1.5*IQR were categorized as outliers and indicated as such on all graphs. This was done to remove bias from extreme outliers observed in this experiment.

## Statistical analysis

Phenotypic analyses were conducted using tissue collected from at least three mice per genotype from at least two independent litters. The number of mice and replicates used for each analysis ('N') are indicated in *Supplementary file 1*. Samples from each mouse were only used for one technical replicate. Power analysis using pilot data was used to determine samples sizes with $\alpha$=0.05 and $\beta$=0.80. Phenotypes were indistinguishable between male and female mice and were analyzed together. Although experimenters were blind to genotype during analysis, in many cases highly penetrant phenotypes revealed the genotypes of the mice and no blinding could be faithfully performed. Unless otherwise stated, no data was excluded from analysis. For comparisons between two groups, significance was determined using a two-tailed Students t-test. For comparisons between more than two groups, significance was determined using a two-way ANOVA with Tukey HSD post-hoc analysis. Statistical significance was set at $\alpha$=0.05 (p<0.05) and data presented as means ± SEM. Individual

statistical tests and specific p-values are reported in *Supplementary file 2*. Statistical analyses and data visualization were performed in R (version 4.0.2).

## Acknowledgements

This work was funded by NIH Grants R01NS091027 (KMW), R01NS126247 (ES) CureCMD (KMW), F31NS120649 (JNJ), F31NS108522 (DSM), P30NS061800 (OHSU ALM), VA I01-BX004938 (ES), Department of Defense W81XWH-18-1-0598 (ES). The contents of this manuscript do not represent the views of the US Department of Veterans Affairs or the US government.

## Additional information

### Funding

| Funder | Grant reference number | Author |
|---|---|---|
| National Institutes of Health | R01NS091027 | Kevin M Wright |
| National Institutes of Health | F31NS120649 | Jennifer N Jahncke |
| National Institutes of Health | F31NS108522 | Daniel S Miller |
| Cure CMD | | Kevin M Wright |
| Muscular Dystrophy Association | https://doi.org/10.55762/mda.1061101.pc.gr.174090 | Kevin M Wright |
| National Institutes of Health | R01NS126247 | Eric Schnell |
| Department of Defense | W81XWH-18-1-0598 | Eric Schnell |
| Veterans Administration | I01-BX004938 | Eric Schnell |

The funders had no role in study design, data collection and interpretation, or the decision to submit the work for publication.

### Author contributions

Jennifer N Jahncke, Conceptualization, Data curation, Formal analysis, Funding acquisition, Investigation, Visualization, Methodology, Writing - original draft, Writing - review and editing; Daniel S Miller, Funding acquisition, Investigation, Methodology, Writing - review and editing; Milana Krush, Investigation, Writing - review and editing; Eric Schnell, Conceptualization, Supervision, Project administration, Writing - review and editing; Kevin M Wright, Conceptualization, Supervision, Funding acquisition, Writing - original draft, Project administration, Writing - review and editing

### Author ORCIDs

Jennifer N Jahncke http://orcid.org/0000-0003-2319-6109
Eric Schnell http://orcid.org/0000-0002-5623-5015
Kevin M Wright http://orcid.org/0000-0001-5094-5270

### Ethics

This study was performed in strict accordance with the recommendations in the Guide for the Care and Use of Laboratory Animals of the National Institutes of Health. All of the animals were handled according to approved institutional animal care and use committee (IACUC) protocols (TR02_IP00000539) and every effort was made to minimize suffering.

Reviewer #1 (Public Review): https://doi.org/10.7554/eLife.87965.3.sa1
Reviewer #2 (Public Review): https://doi.org/10.7554/eLife.87965.3.sa2
Reviewer #3 (Public Review): https://doi.org/10.7554/eLife.87965.3.sa3
Author Response https://doi.org/10.7554/eLife.87965.3.sa4

## Additional files

### Supplementary files
- Supplementary file 1. Experimental N, technical replicates, and sex for experiments.
- Supplementary file 2. Statistical tests and analysis for all data.
- MDAR checklist

### Data availability
All data generated or analyzed during the study are included in the manuscript and supporting files; Ns for experiments are provided in *Supplementary file 1* and statistical tests are included in *Supplementary file 2*.

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
