## [Editor Report · eLife assessment]

These **important** findings will be of interest for the study of dystroglycanopathies and in the general area of axon migration and synapse formation. This work provides **convincing** conclusions about how a range of dystroglycan mutations alter CCK interneuron axonal targeting and synaptic connectivity in the forebrain, and seizure susceptibility.

---

## [Referee Report · Reviewer #1 (Public Review)]

This important study from Jahncke et al. demonstrates inhibitory synaptic defects and elevated seizure susceptibility in multiple models of dystroglycanopathy. A strength of the paper is the use of a wide range of genetic models to disrupt different aspects of dystroglycan protein or glycosylation in forebrain neurons. The authors use a combination of immunohistochemistry and electrophysiology to identify cellular migration, lamination, axonal targeting, synapse formation/function, and seizure phenotypes in forebrain neurons. This is an elegant study with extensive data supporting the conclusions. The role of dystroglycan and the dystrophin glycoprotein complex (DGC) in cellular migration and synapse formation are of broad interest.

A strength of this paper is the use of several transgenic mouse lines with mutations in genes involved in glycosylation of dystroglycan. Knockout of POMT2 abolishes the majority of dystroglycan glycosylation, while point mutations in B4GAT and FKRP presumably produce more minor changes in glycosylation. This is a powerful approach to investigate the role of glycosylation in dystroglycan function.

---

## [Referee Report · Reviewer #2 (Public Review)]

The manuscript by Jahncke and colleagues is centered on the CCK+ synaptic defects that are a consequence of Dystroglycanopathy and/or impaired dystroglycan-related protein function. The authors use conditional mouse models for Dag1 and Pomt2 to ablate their function in mouse forebrain neurons and demonstrate significant impairment of CCK+/CB1R+ interneuron (IN) development in addition to being prone to seizures. Mice lacking the intracellular domain of Dystroglycan have milder defects, but impaired CCK+/CB1R+ IN axon targeting. The authors conclude that the milder dystroglycanopathy is due to the partially reduced glycosylation that occurs in the milder mouse models as opposed to the more severe Pomt2 models. Additionally, the authors postulate that inhibitory synaptic defects and elevated seizure susceptibility are hallmarks of severe dystroglycanopathy and are required for the organization of functional inhibitory synapse assembly.

The manuscript is overall, fairly well-written and the description of the phenotypic impact of disruption of Dystroglycan forebrain neurons (and similar glycosyltransferase pathway proteins) demonstrate impairment in axon targeting and organization. There are some questions with regards to interpretation of some of the results from these conditional mouse models. The study is mostly descriptive, and some validation of subunits of the dystroglycan-glycoprotein complex and laminin interactions would go towards defining the impact of disruption of dystroglycan's function in the brain. The statistics and basic analysis of the manuscript appear to be appropriate and within parameters for a study of this nature. Some clarification between the discrepancies between the Walker Warburg Syndrome (WWS) patient phenotypes and those observed in these conditional mouse models is warranted. This manuscript has the potential to be impactful in the Dystroglycanopathy and general neurobiology fields.

The authors have made significant improvements to address my concerns in this resubmission and the previous critiques of the other reviewers since the prior submission. The work is comprehensive in scope and the statistics are appropriate where required. I believe the conclusions to be valid for this study and I don't have any additional recommendations. I believe this work to be of importance to the Dystroglycanopathy and neurobiology fields.

---

## [Referee Report · Reviewer #3 (Public Review)]

The study presents a systematic analysis of how a range of dystroglycan mutations alter CCK/CB1 axonal targeting and inhibition in hippocampal CA1 and impact seizure susceptibility. The study follows up on prior literature identifying a role for dystroglycan in CCK/CB1 synapse formation. The careful assay includes comparison of 5 distinct dystroglycan mutation types known to be associated with varying degrees of muscular dystrophy phenotypes: a forebrain specific Dag1 knockout in excitatory neurons at 10.5, a forebrain specific knockout of the glycosyltransferase enzyme in excitatory neurons, mice with deletion of the intracellular domain of beta-Dag1 and 2 lines with missense mutations with milder phenotypes. They show that forebrain glutamatergic deletion of Dag1 or glycosyltransferase alters cortical lamination while lamination is preserved in mice with deletion of the intracellular domain or missense mutation. The study extends prior works by identifying that forebrain deletion of Dag1 or glycosyltransferase in excitatory neurons impairs CCK/CB1 and not PV axonal targeting and CB1 basket formation around CA1 pyramidal cells. Mice with deletion of the intracellular domain or missense mutation show

limited reductions in CCK/CB1 fibers in CA1. Carbachol enhancement of CA1 IPSCs was reduced both in forebrain knockouts. Interestingly, carbachol enhancement of CA1 IPSCs was reduced when the intracellular domain of beta-Dag1was deleted, but not I the missense mutations, suggesting a role of the intracellular domain in synapse maintenance. All lines except the missense mutations , showed increased susceptibility to chemically induced behavioral seizures. Together, the study, is carefully designed, well controlled and systematic. The results advance prior findings of the role for dystroglycans in CCK/CB1 innervations of PCs by demonstrating effects of more selective cellular deletions and site specific mutations in extracellular and intracellular domains.

Prior concerns regarding CCK/CB1 cell numbers and potential changes in basal synaptic inhibition are addressed in the revision.

---

## [Author Response]

The following is the authors’ response to the original reviews.

We thank the reviewers for their service and are pleased to see that they were positive about the overall study. The reviewers provided several very good suggestions that we feel have improved the revised manuscript. In response to their suggestions, we have added four new figures of additional data (Figure 1, Supplement 2; Figure 2, Supplement 2; Figure 3, Supplements 1 and 2) in this revision. We have addressed the specific review comments/suggestions point-by-point below. Text changes in the manuscript are indicated in red with line numbers indicated.

**Public Reviews:**

**Reviewer #1 (Public Review):**
This important study from Jahncke et al. demonstrates inhibitory synaptic defects and elevated seizure susceptibility in multiple models of dystroglycanopathy. A strength of the paper is the use of a wide range of genetic models to disrupt different aspects of dystroglycan protein or glycosylation in forebrain neurons. The authors use a combination of immunohistochemistry and electrophysiology to identify cellular migration, lamination, axonal targeting, synapse formation/function, and seizure phenotypes in forebrain neurons. This is an elegant study with extensive data supporting the conclusions. The role of dystroglycan and the dystrophin glycoprotein complex (DGC) in cellular migration and synapse formation are of broad interest.• A strength of this paper is the use of several transgenic mouse lines with mutations in genes involved in glycosylation of dystroglycan. Knockout of POMT2 abolishes the majority of dystroglycan glycosylation, while point mutations in B4GAT and FKRP presumably produce more minor changes in glycosylation. This is a powerful approach to inves5gate the role of glycosylation in dystroglycan function. However, the authors do not address how mutations in these genes may affect glycosylation or expression of proteins other than dystroglycan. It is possible, even likely, that some of the phenotypes observed are due to changing glycosylation in any number of other proteins. The paper would be strengthened by addressing this possibility more directly.

We are glad to see that the reviewer appreciated the range of transgenic models used to define the role of Dag1 glycosylation. It is certainly possible that glycosylation of proteins other than Dag1 is affected by deletion of Pomt2, B4Gat1 and/or FKRP. Indeed, Cadherin and Plexin proteins undergo Omannosylation in the brain. However, recent work has shown that these proteins are not dependent on Pomt1/2 for their O-mannosylation, and use an alternative glycosylation pathway. Therefore, they unlikely to contribute to the phenotypes we observed in our Pomt2, B4Gat1 and/or FKRP mutants. Furthermore, we did not observe any phenotypes in these models that was not also observed in the Dag1 conditional knockouts. We have clarified this point in the results section (lines 117-121) with additional references, and added the caveat that Pomt2, B4gat1, and Fkrp could play a role in the glycosylation of proteins other than Dag1.

• It would be helpful to have a more clear description of how dystroglycan glycosylation is altered in B4GAT1M155T or FKRPP448L mice. For example, Figure 1 makes it appear that the distal sugar moieties are missing, however, the IIH6 antibody, which binds to terminal matriglycan repeats on the glycan chain, recognizes dystroglycan in these mutants.

We apologize for the confusion caused by our schematic in Figure 1. We have adjusted the opacity of the schematic in Figure 1A to better illustrate that the matriglycan chain is s5ll present, albeit at reduced levels, in the B4Gat1 and FKRP mutants. In addition, this is directly shown in the western blot in Figure 1B.

• In Figure 1, the authors use the IIH6 antibody, which recognizes the terminal portion of the dystroglycan glycan chain, to label dystroglycan in the hippocampus. As expected, Emx1Cre,POMT2cKO mice, which lack glycosylation of dystroglycan, do not show any labelling. However, this experiment does not reveal anything about dystroglycan expression, only that the IIH6 antibody no longer recognizes dystroglycan. It would be very helpful in interpreting the later results to know whether the level and pattern of dystroglycan expression is normal or absent in the POMT2cKO mice, perhaps using another antibody that does not target the glycosylated region. For example, figure 3 shows reduced axon targeting to the cell body layer in POMT2cKO, however, it is unclear whether this is due to absence/mislocalization of dystroglycan at the cell surface, or if dystroglycan expression is normal, but glycosylation is directly required for axon targeting.

Addressed in the “Recommendation for Authors” section below

• In Figures 3 and 5, the authors use CB1R labelling to measure axon targeting and synapses formation. However, it is not clear how the authors measure axon targeting and synapses number separately using the same CB1R antibody. In addition, figure 3 shows reduced CB1R labelling in Dag1cyto pyramidal cell layer, but Figure 5 shows no change in CB1R labelling in the same mice. These results would appear to be contradictory.

In Figure 3, the data reflects fluorescent intensity of CB1R+ axons measured across the en5re hippocampal depth. In contrast, the synapse number in Figure 5 is measured as VGat+ and CB1R+ puncta (axonal swellings) within the pyramidal cell layer (SP). The discrepancy between these measurements in the Dag1Cyto mutants likely reflects a change in the distribution of the synaptic contacts in SP (ie: increased contacts in the upper portion of the SP relative to the bottom). This is clarified in the text, lines 315-319.

• The authors measure spontaneous IPSCs (sIPSC) in CA1 pyramidal neurons to measure inhibitory synaptic function. This measure assesses inhibitory synaptic input from all sources, but dystroglycan mutations primarily impairs synapses arising from CCK+/CB1R interneurons, leaving synapses arising from PV or other interneurons relatively unchanged. To assess changes in CCK+/CB1R interneurons the authors apply the cholinergic receptor agonist Carbachol (which selectively activates CCK+/CB1R interneurons) and measure the change in sIPSC amplitude and frequency. While this is an interesting and reasonable experiment, the observed effects could be due to altered carbachol sensitivity in the transgenic mice. Control experiments showing that the effect of Carbachol on excitability of CCK+/CB1R interneurons is similar across mouse lines is missing.

The reviewer is correct that we did not show that CCK/CB1R+ interneurons have the same sensitivity to CCh in controls and the various mutants. Indeed, this is something we have struggled with over the course of the study, and is an inherent limitation of the current study. Unfortunately, these cells are relatively sparse in the CA1, and therefore patching onto presumptive CCK/CB1R+ INs at random to test this directly is not feasible. There are also no genetic or viral tools that we are aware of at this time to fluorescently label these cells for targeted recordings (this would need to be a Cre-independent transgenic mouse line since we are using Cre to delete Dag1 and Pomt2). We tried to assess this by measuring c-fos immunohistochemistry staining as a proxy for activity in response to CCh. Briefly, we incubated acute slices with NBQX, SR95531, and Kynurenic Acid to block synaptic activity, and added CCh in the bath for 30, 60, and 90 minutes to induce CCK/CB1R+ INs firing. Slices were then fixed and stained for c-fos and NECAB1 to identify the CCK/CB1R+ interneurons.

Unfortunately, we had a very difficult time imaging these slices, and we were not confident in our ability to localize c-fos+/NECAB1+ cells. We have clarified that this is an inherent limitation to the study in the text, lines 394-396.

• Earlier work has shown that selective deletion of dystroglycan from pyramidal neurons produces near complete loss of CCK+/CB1R interneurons and synapse formation, a more severe deficit than observed here using a more widespread Cre-driver. This finding is surprising, as generally more wide-spread gene deletion results in more severe, not less severe, phenotypes. The authors make the reasonable claim that more wide-spread gene deletion better mimics human pathologies. However, possible speculation on why this is the case for dystroglycan could provide insight into the nature of CNS deficits in different forms of dystroglycanopathies.

The reviewer is correct that previous work from both our lab and others have shown that deletion of Dag1 from only pyramidal neurons with NEX-cre leads to a complete loss of CCK/CB1R+ INs, and is thus more severe than the phenotype seen with the broader deletion of Dag1 with Emx1-Cre. We were also surprised by this result, so we also generated Dag1;Nestin-Cre mice. These mice show an iden5cal phenotype as the Dag1;Emx1-Cre mutants (new data; Figure 3, Supplement 1; text lines 226-233). This makes us confident in the validity of the Dag1;Emx-Cre mutants with regards to modeling the human disease. We do not know why the NEX-Cre line shows a more severe phenotype; it is possible that this is due to an unknown epistatic interaction between Dag1 and NEX-Cre.

**Reviewer #2 (Public Review):**
The manuscript by Jahncke and colleagues is centered on the CCK+ synaptic defects that are a consequence of Dystroglycanopathy and/or impaired dystroglycan-related protein function. The authors use conditional mouse models for Dag1 and Pomt2 to ablate their function in mouse forebrain neurons and demonstrate significant impairment of CCK+/CB1R+ interneuron (IN) development in addition to being prone to seizures. Mice lacking the intracellular domain of Dystroglycan have milder defects, but impaired CCK+/CB1R+ IN axon targeting. The authors conclude that the milder dystroglycanopathy is due to the par5ally reduced glycosylation that occurs in the milder mouse models as opposed to the more severe Pomt2 models. Additionally, the authors postulate that inhibitory synaptic defects and elevated seizure susceptibility are hallmarks of severe dystroglycanopathy and are required for the organization of functional inhibitory synapse assembly.The manuscript is overall, fairly well-written and the description of the phenotypic impact of disruption of Dystroglycan forebrain neurons (and similar glycosyltransferase pathway proteins) demonstrate impairment in axon targeting and organization.

There are some questions with regards to interpretation of some of the results from these conditional mouse models.

• The study is mostly descriptive, and some validation of subunits of the dystroglycanglycoprotein complex and laminin interactions would go towards defining the impact of disruption of dystroglycan's function in the brain.

Addressed in the “Recommendation for Authors” section below

• The statistics and basic analysis of the manuscript appear to be appropriate and within parameters for a study of this nature.• Some clarification between the discrepancies between the Walker Warburg Syndrome (WWS) patient phenotypes and those observed in these conditional mouse models is warranted. This manuscript has the potential to be impactful in the Dystroglycanopathy and general neurobiology fields.

Addressed in the “Recommendation for Authors” section below

**Reviewer #3 (Public Review):**
The study presents a systematic analysis of how a range of dystroglycan mutations alter CCK/CB1 axonal targeting and inhibition in hippocampal CA1 and impact seizure susceptibility. The study follows up on prior literature identifying a role for dystroglycan in CCK/CB1 synapse formation. The careful assay includes comparison of 5 distinct dystroglycan mutation types known to be associated with varying degrees of muscular dystrophy phenotypes: a forebrain specific Dag1 knockout in excitatory neurons at 10.5, a forebrain specific knockout of the glycosyltransferase enzyme in excitatory neurons, mice with deletion of the intracellular domain of beta-Dag1 and 2 lines with missense mutations with milder phenotypes. They show that forebrain glutamatergic deletion of Dag1 or glycosyltransferase alters cortical lamination while lamination is preserved in mice with deletion of the intracellular domain or missense mutation.

The study extends prior works by identifying that forebrain deletion of Dag1 or glycosyltransferase in excitatory neurons impairs CCK/CB1 and not PV axonal targeting and CB1 basket formation around CA1 pyramidal cells. Mice with deletion of the intracellular domain or missense mutation show limited reductions in CCK/CB1 fibers in CA1. Carbachol enhancement of CA1 IPSCs was reduced both in forebrain knockouts. Interestingly, carbachol enhancement of CA1 IPSCs was reduced when the intracellular domain of beta-Dag1was deleted, but not I the missense mutations, suggesting a role of the intracellular domain in synapse maintenance. All lines except the missense mutations, showed increased susceptibility to chemically induced behavioral seizures. Together, the study, is carefully designed, well controlled and systematic. The results advance prior findings of the role for dystroglycans in CCK/CB1 innervations of PCs by demonstrating effects of more selective cellular deletions and site specific mutations in extracellular and intracellular domains. The interesting finding that deletion of intracellular domain reduces both CB1 terminals in CA1 and carbachol modulation of IPSCs warrants further analysis. Lack of EEG evaluation of seizure latency is a limitation.

Specific comments• Whether CCK/CB1 cell numbers in the CA1 are differentially affected in the transgenic mice is not clarified.

This is a good point; we have now addressed this in Figure 3, Supplement 2 (new data; text lines 234-245). In brief, using two different markers (NECAB1 and NECAB2), we see no change in the number of CCK+/CB1R+ INs in the mutant mice.

• 2. Whether basal synaptic inhibition is altered by the changes in CCK innervation is not examined.

We apologize for the confusion. This is addressed in the text, lines 371-375:

“Notably, even baseline sIPSC frequency was reduced in Dag1cyto/- mutants (2.27±1.70 Hz) compared to WT controls (4.46±2.04 Hz, p = 0.002), whereas baseline sIPSC frequencies appeared normal in all other mutants when compared to their respective controls.”

**Reviewer #1 (Recommendations For The Authors):**
Line 321- CCH-mediated CHANGE in sIPSC amplitude...

This has been corrected (now line 356)

**Reviewer #2 (Recommendations For The Authors):**
Major Comments:• Disruption of the dystroglycan (and subsequent glycosyltransferase proteins) in the brain would likely impact laminin localization and cytoskeletal stability of the dystroglycanprotein complex. The authors should assess (via immunolabeling) the disruption laminin using laminin IF in the various conditional mouse model forebrain sections.

We have stained brains from Dag1, Pomt2, and Dag1cyto mutants with an antibody to Laminin (new data; Figure 2, Supplement 2; text lines 191-205). Briefly, the data clearly shows that laminin staining is abnormal on the pial surface and in the blood vessels of the Dag1;Emx1-cre mutants. This is less severe in the Pomt2;Emx1 mutants, and normal in the Dag1cyto mutants. We also examined higher magnification of laminin staining in hippocampal SP around the pyramidal cells. Laminin in the region was diffuse (not synaptically localized) and there was no difference between any of the mutants and their respective controls (data not shown).

• 2. The biggest question(s) I have is if the synaptic defects that were measured (Fig 6) in the spontaneous inhibitory post-synaptic currents (sIPSCs) could be rescued as a function of the glycosylation of dystroglycan? While ribitol/CDP-ribose has been shown to enhance alpha-dystroglycan glycosylation and total glycosylation, it might be appropriate here. NADplus exogenous supplementation has been (Ortez-Cordero et al., eLife, 2021) has a faster acting effect on glycosylation of dystroglycan and may work in this context. Can the authors add NADplus prior to their CCK+/CB1R+ IN recordings and evaluate synaptic current effects to determine if glycosylation rescue can actually occur?

We are very much interested in the potential to rescue synaptic defects in the various mutants, and this is an active area of study for us going forward. However, we do not think the suggested experiments involving ribitol/NADplus supplementation are likely to work in our specific experiments with these models. In Dag1;Emx1-Cre and Pomt2;Emx1-Cre mice, which show the most dramatic phenotype, there is no O-mannosyl chain ini5ated for ribitol to act upon. In the Dag1Cyto mice, matriglycan is normal and therefore ribitol supplementation is unlikely to have an effect. In B4Gat1 and FKRP mutants, while matriglycan is reduced, there is no significant functional synaptic defect observed. Therefore, even if ribitol was able to increase matriglycan in these two mutants, we would be unable to detect a functional difference. As a side note, while the NADplus supplementation is an interesting idea, the previous study cited did these experiments in vitro in cell lines, so it is not clear if this would have the same effect in vivo. In addition, the time frame that they analyzed was following 24-72 hours of supplementation in cultured cells, which led to ~10% increase in IIH6 at 24 hours. We are unable to incubate acute slices for that amount of time prior to our recordings.

• 3. Minor point. Genetic abbreviation for POMT2 should be "Pomt2", unless some other justification is provided by the authors. I believe the other mutations introduced (e.g. FKRP P448L are humanized mutations).

This has been corrected throughout

• 4. While dystroglycan glycosylation using the IIHC6 antibody is important for proper localization, the core DAG-6F4 monocloncal antibody (DSHB Iowa Hybridoma Bank) would inform you if there is actual disruption in the amount of dystroglycan protein translation and/or production in the forebrain. Can the authors address this question on total dystroglycan production?

This is a great suggestion. We obtained both the DAG-6F4 monoclonal antibody from DSHB and a monoclonal antibody to alpha-Dag1 from Abcam (45-3) and tried using them for immunostaining, but they did not work with brain tissue. However, we were able to use an antibody to beta-Dag1 (Leica, B-DG-CE) for immunostaining. This new data is included in Figure 1, Supplement 2 (text lines 134-140) and shows that as expected, beta-Dag1 is completely gone in Dag1;Emx1-Cre and Dag1Cyto mutants. In the Pomt2;Emx1-Cre mutants, betaDag1 is present but no longer has the punctate appearance consistent with synaptic localization. We have added a section in the discussion expanding on the interpretation of the data, lines 449-462.

• 5. Please comment more on the structural changes in the forebrain and the presence or lack thereof cobblestone (e.g. lissencephaly) in the POMT2 mutant mice (and the other dystroglycanopathy models)? There appears to be some discordance with that and the human Walker Warburg Syndrome (WWS) patients.

The Pomt2;Emx1-cre mutants show a cobblestone phenotype (identical to the Dag1;Emx1-Cre mutants), see Figure 2. This is consistent with these two models having a complete loss of Dag1 function, and therefore modeling the most severe forms of dystroglycanopathy (WWS, MEB). In contrast, the B4Gat1 and FKRP mutants show relatively normal cortical migration because these mutants are hypomorphic and therefore retain some degree of functional Dag1. These two mice model a milder form of dystroglycanopathy. We have clarified this on lines 188-190 and 573-578.

• 6. Line 577. Minor typo, statement ended in a comma, versus a period.

Done

• 7. Methods. Please report on the sex of the mice used in the experiments.

Mice of both sexes were used throughout the study. This has been clarified in the methods section, and we have added information regarding how many mice of each sex were used in each experiment in supplemental table 1

**Reviewer #3 (Recommendations For The Authors):**
Additional Specific Comments,• Although authors include n slice/animals and other details in the methodology, including data as % changes and n (slices/animals) in results will greatly improve the readability.

We have clarified that only one cell per slice was used for physiological recordings (Figure 6) in the methods section, as CCh does not wash out.

• 2. IPSCs are measured as inward currents in high chloride with AMPA blockers which is appropriate. However, Mg was appears to be low (1 mM) in cutting solution. Was this the case in the recording solution. If so, why were NMDA blockers not used.

To clarify, 10mM Mg was included in the cutting solution, and 1mM Mg was included in the recording solution. When the cell is clamped at -70mV, 1mM Mg2+ is sufficient to block NMDA receptors: haps://www.nature.com/ar5cles/309261a0